# MULTI-MARGINAL TEMPORAL SCHRÖDINGER BRIDGE MATCHING FROM UNPAIRED DATA

## ABSTRACT

Many natural dynamic processes -such as in vivo cellular differentiation or disease progression- can only be observed through the lens of static sample snapshots. While challenging, reconstructing their temporal evolution to decipher underlying dynamic properties is of major interest to scientific research. Existing approaches enable data transport along a temporal axis but are poorly scalable in high dimension and require restrictive assumptions to be met. To address these issues, we propose *Multi-Marginal temporal Schrödinger Bridge Matching* (**MMtSBM**) *for video generation from unpaired data*, extending the theoretical guarantees and empirical efficiency of Diffusion Schrödinger Bridge Matching (Shi et al., 2023) by deriving the Iterative Markovian Fitting algorithm to multiple marginals in a novel factorized fashion. Experiments show that MMtSBM retains theoretical properties on toy examples, achieves state-of-the-art performance on real world datasets such as transcriptomic trajectory inference in 100 dimensions, and for the first time recovers couplings and dynamics in very high dimensional image settings, effectively generating temporally coherent videos from purely unpaired data. Our work establishes multi-marginal Schrödinger bridges as a practical and theoretically principled approach for recovering hidden dynamics from static data. code: github.com/ICLRMMtDSBM/MMDSBM_ILCR | website: mmdsbm.notion.site

## 1 INTRODUCTION

The observation of many natural processes yields partial information, resulting in limited time resolution and unpaired snapshots of data. Common examples of this are single-cell sequencing and in vivo biological imaging, where existing methods are destructive and thus cannot link two observations coming from the same cell at different timestamps. The ability to recover the true underlying dynamic from time-unpaired data samples is a key motivation for developing improved methods of trajectory inference.

The modelization of this problem is inherently probabilistic, given both the variability occurring in complex natural processes and the uncertainty of the observation. We thus ask the question: *"What is the most probable evolution of an existing data point, given uncoupled samples of the same process acquired across different times?"*. This point of view has notably been developed in the Schrödinger Bridge (SB) theory (Schrödinger, 1931). The SB is the unique stochastic process whose marginals at start and end times match given probability distributions while minimizing the Kullback–Leibler (KL) divergence w.r.t. a given reference process. The SB also happens to solve a regularized Optimal Transport (OT) problem (Léonard, 2014). Some recent works such as Chen et al. (2019); Lavenant et al. (2024) have explored the theoretical setting of multiple marginals. Recent major advances in statistical learning of SBs have allowed using this framework between complex empirical distributions (De Bortoli et al., 2021; Wang et al., 2021), achieved important improvements in their efficiency (Shi et al., 2023; Bortoli et al., 2024), extended it to the multi-marginal setting and explored various additional constraints such as smooth trajectories (Chen et al., 2023a; Hong et al., 2025), and spline-valued trajectories (Theodoropoulos et al., 2025). A few methods have been proposed to solve the SB problem in an applied machine learning setting. De Bortoli et al. (2021) use iterative proportional fitting (IPF) (Kullback, 1968), the general continuous analogue of the renown Sinkhorn algorithm (Cuturi, 2013). Subsequent works have explored alternative training schemes based on likelihood bounds (Chen et al., 2023b) or on the dual algorithm of IPF: Iterative Markovian Fitting (IMF) (Shi et al., 2023).

A closely related line of work is flow matching (Lipman et al., 2023; Liu et al., 2022; Albergo & Vanden-Eijnden, 2023). These methods have explored OT variants since their inception and have been extended to the multi-marginal setting as well as connected to the Schrödinger Bridge theory (Tong et al., 2024a;b; Kapuśniak et al., 2024).

Concurrent to our work is Park & Lee (2025); we note that they do not scale to video experiments.

Existing multi-marginal methods do not scale to very high dimensions such as image space. Furthermore we believe that existing multi-marginal approaches either make use of modeling assumptions that strongly restrict the class of problems they can solve, such as using spline-valued trajectories, or lack a fully theoretically sound approach.

**Contributions**   This paper makes the following contributions:

1. We define the multi-marginal temporal Schrödinger Bridge problem and demonstrate its fundamental properties (existence and uniqueness of the solution).

2. We introduce a novel factorized extension of the IMF algorithm presented in Shi et al. (2023) to multiple iterative marginals in a way that is efficient –because parallelized along times, and principled –because mathematical sound and with a concrete algorithm closely following theory.

3. We produce a convergence analysis of the algorithm under asymptotic hypotheses.

4. We demonstrate the soundness of the method on low-to-medium-dimensional examples, and achieve state-of-the-art results against comparable methods on 2 widely reported single-cell transcriptomic benchmarks (Moon et al., 2019; Burkhardt et al., 2022).

5. We scale up to 7 iterative marginals in a very high-dimensional *image* setting, presenting for the first time a coherent video generation algorithm from purely time-unpaired data samples.

**Notations**   We adopt the notations from Shi et al. (2023). We denote by $\mathcal{P}(C)$ the space of *path measures*, with $\mathcal{P}(C) = \mathcal{P}(C([0,T], \mathbb{R}^d))$, where $C([0,T], \mathbb{R}^d)$ is the space of continuous functions from $[0,T]$ to $\mathbb{R}^d$. The subset of *Markov path measures* associated with the diffusion $dX_t = v_t(X_t)dt + \sigma_t dB_t$, with $\sigma, v$ locally Lipschitz, is denoted $\mathcal{M}$. We denote $(B_t)_{t \geq 0}$ the $d$-dimensional Brownian motion. For a process $\mathbb{Q}$, the *reciprocal class* of $\mathbb{Q}$ is $\mathcal{R}(\mathbb{Q})$. For $\mathbb{P} \in \mathcal{P}(C)$, we denote by $\mathbb{P}_t$ its marginal at time $t$, by $\mathbb{P}_{s,t}$ the joint law at times $s, t$, and by $\mathbb{P}_{s|t}$ the conditional law at $s$ given $t$. We write $\mathbb{P}_{|t_i,t_j} \in \mathcal{P}(C)$ for the path distribution on $(t_i, t_j)$ given the endpoints $t_i$ and $t_j$; e.g., $\mathbb{Q}_{|t_i,t_j}$ is a scaled Brownian bridge. Unless otherwise specified, $\nabla$ refers to gradients w.r.t. $x_t$ at time $t$. For a joint law $\Pi_{0,T}$ on $\mathbb{R}^d \times \mathbb{R}^d$, the *mixture of bridges measure* is $\Pi = \Pi_{0,T} \mathbb{P}_{|0,T} \in \mathcal{P}(C)$ with $\Pi(\cdot) = \int_{\mathbb{R}^d \times \mathbb{R}^d} \mathbb{P}_{|0,T}(\cdot|x_0, x_T) d\Pi_{0,T}(x_0, x_T)$. The entropy of a process w.r.t. the Brownian motion is denoted $\mathcal{H}$. Finally, for $\pi_0, \pi_T \in \mathcal{P}(X)$, the Kullback–Leibler divergence is $\mathrm{KL}(\pi_0 \| \pi_T) = \int_X \log\left(\frac{d\pi_0}{d\pi_T}(x)\right) d\pi_0(x)$.

## 2 BACKGROUND

### 2.1 THE SCHRÖDINGER BRIDGE PROBLEM

The *Schrödinger Bridge problem* (Schrödinger, 1931) seeks the most likely stochastic evolution between marginals $\mu_0, \mu_T$ under a reference law $\mathbb{Q}$. It admits both a *dynamic* formulation:

$$\mathbb{P}^\star = \underset{\mathbb{P} \in \mathcal{P}(C)}{\arg\min} \, \mathrm{KL}(\mathbb{P} \| \mathbb{Q}) \quad \text{s.t. } \mathbb{P}_0 = \mu_0, \ \mathbb{P}_T = \mu_T, \tag{1}$$

and a *static* formulation on couplings $\Pi \in \mathcal{P}(\mathbb{R}^d \times \mathbb{R}^d)$:

$$\Pi^\star = \underset{\Pi}{\arg\min} \, \mathrm{KL}(\Pi \| \mathbb{Q}_{0,T}) \quad \text{s.t. } \Pi_0 = \mu_0, \ \Pi_T = \mu_T. \tag{2}$$

**Note: Connection to Quadratic OT.**   If $\mathbb{Q}$ is Brownian motion, equation 2 is precisely entropy-regularized quadratic OT with cost $c(x_0, x_T) = \frac{1}{2}\|x_0 - x_T\|^2$ and regularization $\varepsilon = \sigma^2$. In the limit $\varepsilon \to 0$, this recovers classical OT, which motivates our interpolation framework.

## 2.2 Iterative Markovian Fitting (IMF)

The SB solution is the unique path measure that is both *Markovian* and belongs to the *reciprocal class* of $\mathbb{Q}$ while matching marginals (Léonard, 2014). This motivates the *Iterative Markovian Fitting* (IMF) algorithm (Shi et al., 2023; Peluchetti, 2023), which alternates between reciprocal and Markov projections:

$$\mathbb{P}^{2n+1} = \mathrm{proj}_{\mathcal{M}}(\mathbb{P}^{2n}), \qquad \mathbb{P}^{2n+2} = \mathrm{proj}_{\mathcal{R}(\mathbb{Q})}(\mathbb{P}^{2n+1}) \tag{3}$$

These projections admit KL variational characterisations (A.1) and the iterations converge to $\mathbb{P}^{\star}$.

In practice, IMF is implemented by learning the drift of the Markovian projection via a bridge-matching loss (see A.1). Compared to Iterative Proportional Fitting (IPF), IMF preserves both marginals simultaneously and is more efficient (details in A.1).

## 3 Multi-Marginal temporal Schrödinger Bridge Matching

All proofs can be found in the Appendix A.6.

### 3.1 Multi-Marginal temporal Schrödinger Bridge Problem

In the present work, we considered the time-ordered Multi-Marginal Schrödinger Bridge, where the marginals are associated with an underlying temporal axis. In this setting, the goal is not simply to fit an arbitrary number of marginals, but to recover the law of a stochastic process that evolves consistently over time.

Let $0 = t_0 < t_1 < \cdots < t_K = T$ be a fixed time grid, and let $\mu_0, ..., \mu_k, ..., \mu_T \in \mathcal{P}(\mathbb{R}^d)$ denote prescribed marginals at times $(t_k)_{k=0,...,K}$, assuming $\mu_{t_k} \ll Q_{t_k}$ for all $k$. Given a reference process $\mathbb{Q}$ on $C([0,T], \mathbb{R}^d)$, the multi-marginal Schrödinger Bridge problem (MMSB) is defined as

$$\mathbb{P}^{\star} = \underset{\mathbb{P} \in \mathcal{P}(C)}{\mathrm{argmin}} \, \mathrm{KL}(\mathbb{P} \,\|\, \mathbb{Q}) \quad \text{subject to} \quad X_{t_k} \sim \mu_k, \;\; k = 0, \ldots, K \tag{4}$$

**Note: Connection to multi-marginal Optimal Transport** If $\mathbb{Q}$ is associated with a Brownian motion, the induced reference coupling $\mathbb{Q}_{t_0,...,t_N}$ is characterized by independent Gaussian increments $X_{t_{i+1}} - X_{t_i} \sim \mathcal{N}(0, \sigma^2(t_{i+1} - t_i))$. By evaluating the KL term, 4 can be rewritten as:

$$\Pi^{\star} = \arg \min_{\Pi \in \mathcal{P}((\mathbb{R}^d)^N)} \left\{ \mathbb{E}_{X \sim \Pi} \left[ \sum_{i=0}^{K-1} \frac{1}{t_{i+1} - t_i} \|X_{t_{i+1}} - X_{t_i}\|^2 \right] - 2\sigma^2 T \mathcal{H}(\Pi) \;\; : \;\; \Pi_i = \mu_{t_i}, \, \forall i \right\}$$

This is precisely an entropy-regularised multi-marginal OT problem with a time-structured quadratic cost $c(x_0, \ldots, x_N) = \sum_{i=0}^{K-1} \frac{1}{t_{i+1}-t_i} \|x_{i+1} - x_i\|^2$ and entropy-regularisation parameter $\varepsilon = 2\sigma^2$.

This formulation is particularly interesting when no better prior is available, and because of the clear interpretation it allows: when using a Brownian motion as prior, we are approaching quadratic OT. Note however that we do not rely on this assumption at all for theoretical results.

**Classical properties of the multi-marginal temporal Schrödinger bridge** We first demonstrate a set of classical properties that characterize MMSB (4) and guide the construction of our method.

**Definition 3.1** (Static formulation). *Let $\mathbb{Q}_{t_0,...,t_K}$ be the joint law of $\mathbb{Q}$ at $0 = t_0 < \cdots < t_K = T$. The static problem is*

$$\pi^{\star} = \arg \min_{\pi \in \Pi(\pi_{t_0}, ..., \pi_{t_K})} KL(\pi \,\|\, \mathbb{Q}_{t_0,...,t_K}),$$

*where $\Pi(\pi_{t_0}, \ldots, \pi_{t_K})$ denotes couplings on $(\mathbb{R}^d)^{K+1}$ with marginals $\pi_{t_i}$.*

The MMSB is therefore a projection of the reference law onto the set of couplings with prescribed marginals. The following results ensure that this problem is well posed and that the solution has a convenient structure.

**Proposition 3.1** (Existence and uniqueness). *The MMSB admits a unique solution $P^{\star}$.*

This guarantees that the iterative algorithms we design later target a well-defined object. Moreover, the solution can be described equivalently in both static and dynamic terms.

**Proposition 3.2** (Dynamic–static equivalence)**.** *The dynamic solution $P^\star$ is determined by the static one $\pi^\star$:*

$$\pi^\star = P^\star_{t_0,\ldots,t_K}, \qquad P^\star = \pi^\star \otimes \mathbb{Q}(\cdot \mid X_{t_0}, \ldots, X_{t_K}).$$

This equivalence highlights that solving the static problem is enough to recover the full path measure. In addition, the structure of $\mathbb{Q}$ plays a key role in the nature of the solution.

**Proposition 3.3** (Markovianity)**.** *If $\mathbb{Q}$ is Markov, then the MMSB solution $P^\star$ is Markov.*

These properties ensure that we can restrict our search to Markovian (and therefore reciprocal A.1) measures, which will be central to the projection algorithms introduced later. Finally, the explicit form of the solution further clarifies its structure.

**Proposition 3.4** (Form of the solution)**.** *Under mild assumptions:*

$$P^\star = \pi^\star \otimes \mathbb{Q}(\cdot \mid X_{t_0}, \ldots, X_{t_K}), \qquad \frac{d\pi^\star}{d\mathbb{Q}_{t_0,\ldots,t_K}}(x_0, \ldots, x_K) = \prod_{i=0}^{K} f_i(x_i).$$

where the $f_i$'s are functions of the Lagrange multipliers for the marginal constraints (see A.6.6). This factorized form motivates the use of alternating projections and parametric families of potentials in the iterative algorithm that we develop in the next section.

### 3.2 ITERATIVE MARKOVIAN FITTING FOR MULTI MARGINAL TEMPORAL SCHRÖDINGER BRIDGE

#### 3.2.1 MULTI-MARGINAL MARKOV AND RECIPROCAL PROJECTIONS

To construct an algorithm for MMSB, we first extend the notions of reciprocal and Markovian projections to the multi-marginal setting. The idea is to approximate the global bridge by a sequence of independent sub-bridges, and to alternate between reciprocal and Markovian structures.

**Definition 3.2** (Factorized reciprocal class and projection)**.** *For each interval $[t_i, t_{i+1}]$ and endpoints $(x_i, x_{i+1})$, let $\mathbb{Q}^{x_i,x_{i+1}}_{[t_i,t_{i+1}]}$ denote the bridge of $\mathbb{Q}$ between $x_i$ and $x_{i+1}$. Given a coupling $\pi$ on $(\mathbb{R}^d)^{K+1}$, define*

$$P = \int \bigotimes_{i=0}^{K-1} \mathbb{Q}^{x_i,x_{i+1}}_{[t_i,t_{i+1}]} \, \pi(dx_0, \ldots, dx_K).$$

*The factorized reciprocal class, denoted $\mathcal{R}^{\otimes}(\mathbb{Q})$, is the set of all such measures $P$.*

*Moreover, for any $P \in \mathcal{P}(C([0,T], \mathbb{R}^d))$, the reciprocal projection onto $\mathcal{R}^{\otimes}(\mathbb{Q})$ is defined as*

$$\Pi^\star = \mathrm{proj}_{\mathcal{R}^{\otimes}(\mathbb{Q})}(P) = P_{t_0,\ldots,t_K} \bigotimes_{i=0}^{K-1} \mathbb{Q}^{x_i,x_{i+1}}_{[t_i,t_{i+1}]},$$

*i.e. we keep the marginals $P_{t_0,\ldots,t_K}$ at the grid points and fill the dynamics between them with independent bridges of $\mathbb{Q}$ conditioned on the endpoints $(x_i, x_{i+1})$.*

*Equivalently, $\Pi^\star$ admits the variational characterization*

$$\Pi^\star = \arg\min_{\Pi \in \mathcal{R}^{\otimes}(\mathbb{Q})} KL(P \,\|\, \Pi).$$

**Proposition 3.5** (Local reciprocal structure of the factorized class)**.** *Let $\mathbb{Q}$ be a reference Markov process and let $P \in \mathcal{R}^{\otimes}(\mathbb{Q})$ belong to the factorized reciprocal class. Then, for each subinterval $[t_{i-1}, t_i]$, the restriction of $P$ to $C([t_{i-1}, t_i], \mathbb{R}^d)$ is in the reciprocal class of $\mathbb{Q}$ over $[t_{i-1}, t_i]$. In particular, conditionally on the endpoints $(X_{t_{i-1}}, X_{t_i})$, the law of $P$ coincides with the bridge of $\mathbb{Q}$ between $t_{i-1}$ and $t_i$.*

This class provides a tractable approximation: each sub-interval is filled with the bridge of $\mathbb{Q}$, while the global coupling ensures consistency across marginals. Hence, factorized bridges inherit local reciprocity, which justifies their use as a relaxation of the true reciprocal class.

This projection enforces the prescribed marginals while completing the dynamics with local bridges. In contrast, the Markovian projection seeks a single Markov diffusion with consistent marginals.

**Definition 3.3** (Markovian projection in the factorized setting). *Let $\Pi$ be the factorized mixture of independent Brownian bridges. For any $t \in [0, T]$, let $i(t)$ be the unique index such that $t \in [t_{i(t)}, t_{i(t)+1}]$. We employ a slight abuse of notation and subsequently write $i$ instead of $i(t)$.*

*The Markovian projection of $\Pi$, denoted $M^\star = \mathrm{proj}_{\mathcal{M}}(\Pi)$, is the unique diffusion process*

$$dX_t^\star = \left\{ f_t(X_t^\star) + v_t^\star(X_t^\star) \right\} dt + \sigma_t \, dB_t,$$

*with effective drift*

$$v_t^\star(x) = \sigma_t^2 \, \mathbb{E}_{\Pi_{t_{i+1}|t}} \left[ \nabla \log \mathbb{Q}_t^{|t_i, t_{i+1}}(X_{t_{i+1}} \mid X_t) \,\Big|\, X_t = x \right] \stackrel{Brownian}{=} \frac{\mathbb{E}_{\Pi_{t_{i+1}|t}}[X_{t_{i+1}} \mid X_t = x] - x}{t_{i+1} - t}$$

*By the Markovian projection theorem of Gyöngy (1986), and as further developed in Peluchetti (2023); De Bortoli et al. (2021), the process $M^\star$ is Markov and matches the one-dimensional marginals of the original factorized law $\Pi$.*

**Proposition 3.6** (Variational characterization of the factorized Markovian projection). *Assume that $\sigma_t > 0$. Let $M^\star = \mathrm{proj}_{\mathcal{M}}(\Pi)$ be the Markovian projection of $\Pi$ as in Definition 3.3. Then:*

$$M^\star = \arg \min_{M \in \mathcal{M}} \left\{ KL(\Pi \,\|\, M) \right\},$$

*and*

$$KL(\Pi \,\|\, M^\star) = \frac{1}{2} \int_0^T \mathbb{E}_{\Pi_{t_i, t}} \left[ \frac{1}{\sigma_t^2} \left\| \sigma_t^2 \, \mathbb{E}_{\Pi_{t_{i+1}|t}} \left[ \nabla \log \mathbb{Q}_t^{|t_i, t_{i+1}}(X_{t_{i+1}} \mid X_t) \,\big|\, X_t, X_{t_i} \right] - v_t^\star(X_t) \right\|^2 \right] dt$$

*In addition, for any $t \in [0, T]$, the time marginal of $M^\star$ coincides with that of $\Pi$: $M_t^\star = \Pi_t$.*
*In particular, $M_{t_i}^\star = \Pi_{t_i}$ for all grid points $t_i$.*

Together, these results allow us to alternate between reciprocal and Markovian structures in the multi-marginal setting. Importantly, the Markovian projection admits explicit forward and backward formulations.

**Proposition 3.7.** *Let $\Pi \in \mathcal{R}^{\otimes}(\mathbb{Q})$. Under mild regularity conditions, the Markovian projection $M^\star = \mathrm{proj}_{\mathcal{M}}(\Pi)$ is associated with the forward SDE*

$$dX_t = \left\{ f_t(X_t) + \sigma_t^2 \, \mathbb{E}_{\Pi_{t_{i+1}|t}} \left[ \nabla \log \mathbb{Q}_t^{[t_i, t_{i+1}]}(X_{t_{i+1}} \mid X_t) \,\big|\, X_t \right] \right\} dt + \sigma_t dB_t, \quad X_{t_i} \sim \mu_{t_i} \quad (5)$$

*and with the backward SDE*

$$dY_t = \left\{ -f_{t_{i+1}-t}(Y_t) + \sigma_{t_{i+1}-t}^2 \, \mathbb{E}_{\Pi_{t_i|t}} \left[ \nabla \log \mathbb{Q}_t^{[t_i, t_{i+1}]}(Y_{t_i} \mid Y_t) \,\big|\, Y_t \right] \right\} dt + \sigma_{t_{i+1}-t} dB_t, \quad Y_{t_{i+1}} \sim \mu_{t_{i+1}} \quad (6)$$

This key result highlights that the Markovian projection can be expressed both in the forward and in the backward direction, allowing us to design an algorithm that jointly leverage both dynamics.

**Conjecture 3.1** (Analogue of Léonard (2014) Theorem 2.12). *Let $\mathbb{Q}$ be a Markov reference process. Suppose that $P$ is a Markov path measure such that*

$$P \in \mathcal{R}(\mathbb{Q}), \qquad P_{t_i} = \mu_{t_i}, \quad i = 0, \dots, K.$$

*Then $P$ coincides with the unique solution $P^\star$ of the multi-marginal Schrödinger bridge problem (MMSB) with reference $\mathbb{Q}$.*

### 3.2.2 ITERATIVE MARKOVIAN FACTORIZED FITTING

Based on Conjecture 3.1, we propose a novel algorithm called *Iterative Markovian Factorized Fitting* (IMFF) to solve multi-marginal Schrödinger Bridges. We consider a sequence $(\mathbb{P}^n)_{n \in \mathbb{N}}$ such that

$$\mathbb{P}^{2n+1} = \mathrm{proj}_{\mathcal{M}}(\mathbb{P}^{2n}), \qquad \mathbb{P}^{2n+2} = \mathrm{proj}_{\mathcal{R}^{\otimes}(\mathbb{Q})}(\mathbb{P}^{2n+1}), \quad (7)$$

with $\mathbb{P}^0$ such that $\mathbb{P}_{t_i}^0 = \mu_{t_i}$ for all $i = 0, \dots, K$, and $\mathbb{P}^0 \in \mathcal{R}^{\otimes}(\mathbb{Q})$. These updates correspond to alternatively performing Markovian projections and factorized reciprocal projections in order to enforce all prescribed marginals.

**Lemma 3.1** (Pythagorean identities in the factorized setting). *Under mild assumptions, if $M \in \mathcal{M}$, $\Pi \in \mathcal{R}^{\otimes}(\mathbb{Q})$ and $KL(\Pi \| M) < +\infty$, we have*

$$KL(\Pi \| M) = KL(\Pi \| \operatorname{proj}_{\mathcal{M}}(\Pi)) + KL(\operatorname{proj}_{\mathcal{M}}(\Pi) \| M)$$

*Similarly, if $KL(M \| \Pi) < +\infty$, we have*

$$KL(M \| \Pi) = KL(M \| \operatorname{proj}_{\mathcal{R}^{\otimes}(\mathbb{Q})}(M)) + KL(\operatorname{proj}_{\mathcal{R}^{\otimes}(\mathbb{Q})}(M) \| \Pi)$$

**Proposition 3.8.** *Under mild assumptions, we have*

$$KL(P^{n+1} \| P^{\star}) \leq KL(P^n \| P^{\star}) < \infty \quad and \quad \lim_{n \to \infty} KL(P^n \| P^{\star}) = 0$$

Hence, for the IMFF sequence $(\mathbb{P}^n)_{n \in \mathbb{N}}$, the Markov path measures $(\mathbb{P}^{2n+1})_{n \in \mathbb{N}}$ are getting closer to the factorized reciprocal class $\mathcal{R}^{\otimes}(\mathbb{Q})$, while the reciprocal path measures $(\mathbb{P}^{2n+2})_{n \in \mathbb{N}}$ are getting closer to the set of Markov measures. This mirrors the situation in the classical IMF setting, but now in the multi-marginal framework.

**Theorem 3.2.** *Under mild assumptions, the IMFF sequence $(\mathbb{P}^n)_{n \in \mathbb{N}}$ admits at least one fixed point $\mathbb{P}^{\star}$, and we have:*

$$\lim_{n \to +\infty} KL(\mathbb{P}^n \| \mathbb{P}^{\star}) = 0$$

*Moreover, denoting by $\mathbb{P}^{\text{MMSB}}$ the solution of ([MMSB](#)) and by $\mathbb{P}^{\text{pair}}$ the collage of pairwise Schrödinger Bridges, the limit of the IMFF sequence satisfies the inequality:*

$$KL(\mathbb{P}^{\text{MMSB}} \| \mathbb{Q}) = KL(\mathbb{P}^{\star} \| \mathbb{Q}) \leq KL(\mathbb{P}^{\text{pair}} \| \mathbb{Q})$$

*where $\mathbb{Q}$ is the chosen reference process. Thus, $\mathbb{P}^{\star}$ is the multi-marginal Schrödinger Bridge.*

### 3.2.3 THEORETICAL ALGORITHM

The Markovian projection necessitates learning one neural drifts per direction. Concretely, we solve

$$\theta^{\star} = \arg\min_{\theta} \ \mathbb{E}_{\text{batch}} \Big[ \big\| v_{\theta}(X_t, t) - \sigma_t^2 \, \mathbb{E}\big[ \nabla \log \mathbb{Q}_t^{[t_{i(t)}, t_{i(t)+1}]}(X_{t_{i(t)+1}} \mid X_t) \mid X_t \big] \big\|^2 \Big] \quad (8)$$

for the *forward* drift $v_{\theta}$, and

$$\phi^{\star} = \arg\min_{\phi} \ \mathbb{E}_{\text{batch}} \Big[ \big\| v_{\phi}(Y_t, t) - \sigma_{t_{i(t)+1}-t}^2 \, \mathbb{E}\big[ \nabla \log \mathbb{Q}_t^{[t_{i(t)}, t_{i(t)+1}]}(Y_{t_{i(t)}} \mid Y_t) \mid Y_t \big] \big\|^2 \Big] \quad (9)$$

for the *backward* drift $v_{\phi}$.

We summarize in Algorithm 1 our method and provide a practical implementation of IMFF in A.3.

---

**Algorithm 1** Iterative Markovian Factorized Fitting (IMFF)

---

1: **Input:** time grid $0 = t_0 < \cdots < t_K = T$, marginals $(\mu_{t_i})_{i=0}^{K}$, reference process $\mathbb{Q}$, number of iterations $N$
2: **Init:** choose $\mathbb{P}^0 \in \mathcal{R}^{\otimes}(Q)$ with $\mathbb{P}^0_{t_i} = \mu_{t_i}$ for all $i$
3: **for** $n = 0, \ldots, N-1$ **do**
4:     **Backward Markovian step:** learn drift $v_{\phi}$ via SDE equation 6, yielding $\mathbb{P}^{2n+1}$ with $t_i$ updated and $t_{i+1}$ fixed from $(\mu_{t_{i+1}})$.
5:     **Forward reciprocal projection:** $\mathbb{P}^{2n+1} \leftarrow \operatorname{proj}_{\mathcal{R}^{\otimes}(Q)}(\mathbb{P}^{2n+1})$ (cf. Def. 3.2), filling bridges with $Q$ using $t_i$ from $\mathbb{P}^{2n+1}$ and $t_{i+1}$ from the dataset.
6:     **Forward Markovian step:** learn drift $v_{\theta}$ via SDE equation 5, yielding $\mathbb{P}^{2n+2}$ with $t_{i+1}$ updated and $t_i$ fixed from $(\mu_{t_i})$.
7:     **Backward reciprocal projection:** $\mathbb{P}^{2n+2} \leftarrow \operatorname{proj}_{\mathcal{R}^{\otimes}(Q)}(\mathbb{P}^{2n+2})$ (cf. Def. 3.2), filling bridges with $Q$ using $t_{i+1}$ from $\mathbb{P}^{2n+2}$ and $t_i$ from the dataset.
8: **end for**
9: **Output:** learned drifts $(v_{\phi}, v_{\theta})$

---

**Proposition 3.9.** *Suppose the families of functions $\{v_{\theta} : \theta \in \Theta\}$ and $\{v_{\phi} : \phi \in \Phi\}$ are rich enough to represent the optimal forward and backward drifts. Let $(P^n, M^n)_{n \in \mathbb{N}}$ be the sequence produced by Algorithm 1. Then, as $n \to \infty$, we have convergence towards an approximate multi-marginal Schrödinger bridge. Moreover, the Markov law $M^n$ coincides in the limit with the intermediate approximate MMSB solution lying between the true multi-marginal Schrödinger bridge and the pairwise construction.*

## 4 EXPERIMENTS

For all experiments, we employ Brownian motion $(\sigma_t B_t)_{0 \leq t \leq T}$ for the reference measure $\mathbb{Q}$ and $T = N - 1$ where $N$ is the number of marginals.[1] All trainings start after a warmup phase like in Shi et al. (2023), detailed in A.3.

### 4.1 MMtSBM RECOVERS THE EXACT OT BETWEEN GAUSSIAN MIXTURES

In this 2D experiment akin to Liu et al. (2022), we used $N = 3$ mixtures of two standard Gaussian as marginals. In this configuration the optimal transport *between each pair of marginals* can be computed exactly: it is a pure translation of each Gaussian components inside the mixtures, as we verified with POT (see Figure 6). After only the warm-up phase (akin to flow matching (Lipman et al., 2023)), we can see that the learned transport maps *mix* the Gaussian components of the mixtures, resulting in intersecting trajectories as can be seen in the top row of Figure 1. However, *after* the SB learning phase of MMtSBM, we can see in the bottom row that the learned trajectories do *not* intersect each other anymore and that MMtSBM yields the expected exact optimal transport map: pure translations between Gaussian components. This observation is consistent with the theory: the warm-up phase preserves only the Markov property, while the final learned coupling additionally also preserves the reciprocal property, thus corresponding to the true SB. We empirically observe that the optimality emerges gradually along MMtSBM training epochs: trajectories get *rectified* from epoch 1, become optimal around epoch 5, and consistently remain so after. We will now confirm these visual findings with quantitative metrics in 4.2.

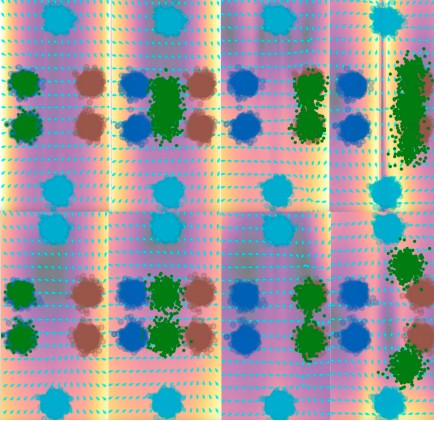

Figure 1: Top row: epoch 0 (only noisy flow matching). Bottom row: epoch 5 (after MMtSBM training). From left to right: snapshots at times $(0, 0.5, 1, 1.3)$. True marginal times $(t_0, t_1, t_2) = (0, 1, 2)$. The order of the 3 true marginals is: $t_0 =$ dark blue; $t_1 =$ red; $t_2 =$ light blue. Generated samples are in green. In the background is the quiver plot of the learned score network.

### 4.2 MMtSBM ACHIEVES GOOD USUAL SB METRICS

To quantitatively verify that MMtSBM recovers the correct multi-marginal SB in terms of both 1) static coupling and 2) energy minimization, we extended the now classical "Moons" and "8Gaussians" experiments found in Tong et al. (2024a) and Shi et al. (2023) to our temporal multi-marginal setting in Table 1 (see Figure 7). Choosing $N = 4$, we considered ($\mathcal{N} \rightarrow$ Moons $\rightarrow \mathcal{N} \rightarrow$ Moons), and ($\mathcal{N} \rightarrow$ 8Gaussians $\rightarrow \mathcal{N} \rightarrow$ 8Gaussians). To assess 1) we report the $\mathcal{W}_2$ distance of generations vs test set data at target marginal time(s), averaging along the $N - 1 = 3$ target times for MMtSBM and comparing this to the single bridge setting. To assess 2) we report the full path energy $\mathbb{E}\left[\int_0^T \|v(t, \mathbf{Z}_t)\|^2 \, dt\right]$ where $Z_t$ is the process simulated along the ODE drift 10.

| | Model | $\mathcal{W}_2$ | Path Energy |
|---|---|---|---|
| **Moons** | Single bridge | $0.144_{\pm 0.024}$ | $1.580_{\pm 0.036}$ |
| | Single bridge $\times 3$ | – | $4.740$ |
| | MMtSBM (ours) | $0.148_{\pm 0.041}$ | $5.350_{\pm 0.085}$ |
| **$8\mathcal{N}$** | Single bridge | $0.338_{\pm 0.091}$ | $14.810_{\pm 0.255}$ |
| | Single bridge $\times 3$ | – | $44.430$ |
| | MMtSBM (ours) | $0.352_{\pm 0.084}$ | $46.920_{\pm 0.285}$ |

Table 1: Comparison in terms of static coupling ("$\mathcal{W}_2$") and energy minimization ("Path Energy"). The rows marked "$\times 3$" correspond to the hypothetical case where the energy of a single bridge is simply tripled, and are included as an ideal baseline for comparison with our actual multi-bridge setting. All metrics apart from ours are from Shi et al. (2023).

We observe that despite a much more complex *time-varying* true transport map to be learned, MMtSBM achieves almost as low $\mathcal{W}_2$ distances than the simple single-bridge setting (3% to 4%), and that our full path energy is within 13% to 6% of the ideal extrapolation of the single bridge result. This validates that MMtSBM manages to approach the true SB in practice.

---

[1]Videos for most experiments can be found at mmdsbm.notion.site.

### 4.3 MMtSBM scales to 50d Gaussian transport

We next proceed to scaling our method to dimension $d = 50$. We follow the setting of (Shi et al., 2023) and consider a Gaussian-to-Gaussian transport experiment, extended to our multi-marginal case. Specifically, we prescribe four Gaussian marginals at times $t = 0, 1, 2, 3$: $\mu_0 = \mathcal{N}(-0.1 \cdot \mathbf{1}_d, I_d), \mu_1 = \mathcal{N}(0.1 \cdot \mathbf{1}_d, I_d), \mu_2 = \mathcal{N}(-0.1 \cdot \mathbf{1}_d, I_d), \mu_3 = \mathcal{N}(0.1 \cdot \mathbf{1}_d, I_d)$ where $\mathbf{1}_d \in \mathbb{R}^d$ denotes the vector of all ones, and $I_d$ is the $d \times d$ identity matrix.

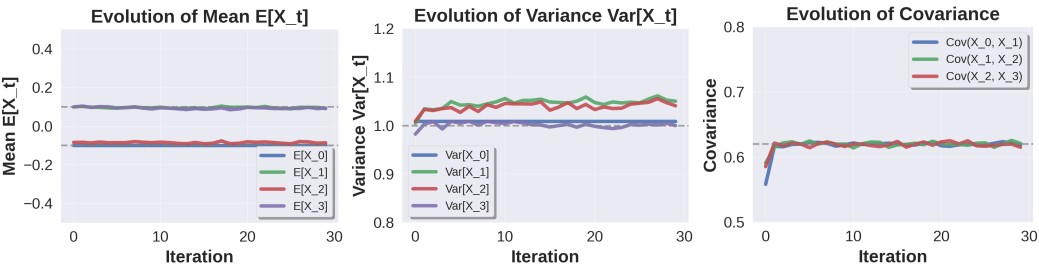

Figure 2: Evolution of mean, variance, and covariance in the multi-marginal $50d$ Gaussian transport. Dash lines are the theoretical true values.

Since no closed-form solution is available for the static multi-marginal SB, we compare our method to the sequence of theoretical results for each *pairwise* SB (Bunne et al., 2023). As shown in Figure 2, the mean converges rapidly to the prescribed values ($0.1$ or $-0.1$) across all four marginals. The variance is slightly more difficult to match: for interior marginals the process tends to overestimate the standard deviation. In contrast, the covariance is consistently well reproduced by our method and remains stable across all three transitions. Interestingly, the covariance converges only after the warmup stage, confirming the added value of the subsequent OT phases. Overall, these results show that MMtSBM scales effectively to the multi-marginal Gaussian setting in $d = 50$.

### 4.4 MMtSBM achieves SOTA results on 100d transcriptomic benchmarks

We next evaluate our method on the the Embryoid Body (EB) (Moon et al., 2019) and MULTI (Lance et al., 2022) benchmarks, two trajectory inference tasks on real single-cell RNA-seq data. We project RNA counts to their first $d = 100$ principal components for each of the $N = 5$ and $N = 4$ marginals, respectively.[2] We report the Maximum Mean Discrepency (MMD) and Sliced Wasserstein Distance (SWD) for EB in Table 2, and the Wasserstein-1 distance for MULTI in Table 3. For the EB benchmark we train on all marginals, while for the MULTI benchmark we leave-out one of either intermediate times ($t = 1$ or $t = 2$) during training.

| | DMSB (Chen et al., 2023a) | | MMtSBM (ours) | |
| Time | MMD ↓ | SWD ↓ | MMD ↓ | SWD ↓ |
| --- | --- | --- | --- | --- |
| $t_1$ | 0.021 | 0.114 | **0.016** | **0.104** |
| $t_2$ | 0.029 | 0.155 | **0.020** | **0.139** |
| $t_3$ | 0.038 | 0.190 | **0.020** | **0.127** |
| $t_4$ | 0.034 | 0.155 | **0.020** | **0.143** |
| Average | 0.032 $_{\pm 3e-3}$ | 0.160 $_{\pm 2e-2}$ | **0.019** $_{\pm 4e-4}$ | **0.130** $_{\pm 2e-3}$ |

| Algorithm | MMD ↓ | SWD ↓ |
| --- | --- | --- |
| NLSB (Koshizuka & Sato, 2023) | 0.66 | 0.54 |
| MIOFlow (Huguet et al., 2022) | 0.23 | 0.35 |
| DMSB (Chen et al., 2023a) | 0.032 $_{\pm 3e-3}$ | 0.16 $_{\pm 2e-2}$ |
| **MMtSBM** (ours) | **0.019** $_{\pm 4e-4}$ | **0.130** $_{\pm 2e-3}$ |

Table 2: MMD and SWD of generations vs test set for the $d = 100$ EB benchmark. Our generations start from $\mu_{t=0}^{\text{test}}$. Top table: per-marginal metrics. Bottom table: average over all marginals. Others' results are from Chen et al. (2023a). Our error margins are over 10 evaluations while DMSB's are over 3. Best value in **bold**.

---

[2]We actually reuse preprocessed data from Tong et al. (2020) and Tong et al. (2024b).

| Method | | $\mathcal{W}_1 (\downarrow)$ | Method | | $\mathcal{W}_1 (\downarrow)$ |
|---|---|---|---|---|---|
| | *Schrödinger Bridge* | | | *Wasserstein Gradient Flows* | |
| WLF-SB | (Neklyudov et al., 2024) | $55.065 \pm 5.499$ | WLF-SB | (Neklyudov et al., 2024) | $55.065 \pm 5.499$ |
| [SF]²M-Exact | (Tong et al., 2024b) | $52.888 \pm 1.986$ | WLF-OT | (Neklyudov et al., 2024) | $55.416 \pm 6.097$ |
| [SF]²M-Geo | (Tong et al., 2024b) | $52.203 \pm 1.957$ | WLF-UOT | (Neklyudov et al., 2024) | $54.222 \pm 5.827$ |
| MMtSBM | (ours) | $\underline{44.542} \pm 0.637$ | WLF-(OT+potential) | (Neklyudov et al., 2024) | $47.365 \pm 0.051$ |
| | | | WLF-(UOT+potential) | (Neklyudov et al., 2024) | $45.231 \pm 0.010$ |
| | *No precomputed OT conditioning* | | | | |
| I-CFM | (Tong et al., 2024a) | $57.262 \pm 3.855$ | | *Flow Matching with exact OT conditioning* | |
| I-MFM$_{RBF}$ | (Kapuśniak et al., 2024) | $54.197 \pm 1.408$ | OT-CFM | (Tong et al., 2024a) | $54.814 \pm 5.858$ |
| MMtSBM | (ours) | $\underline{44.542} \pm 0.637$ | OT-MFM$_{RBF}$ | (Kapuśniak et al., 2024) | $50.906 \pm 4.627$ |
| | | | | *Metric-aware interpolation with exact OT conditioning* | |
| | | | **GAGA** | (Sun et al., 2025) | $\mathbf{27.04} \pm \mathbf{2.95}$ |

Table 3: $\mathcal{W}_1$ of generations vs left-out test set for the $d = 100$ MULTI benchmark. Generations start from $\mu_{i-1}^{\text{test}}$ where $i$ is the left-out time. Reported figures are the average between left-out $t = 1$ and $t = 2$ marginals. Our error margin is over 3 training runs. Best value in **bold**, second best underlined. See A.5.5 for details & comments.

On the EB benchmark, our method consistently outperforms baselines on all marginals, reducing the average MMD from $0.032 \pm 3e-3$ to $\mathbf{0.019} \pm 4e-4$ and the SWD from $0.16 \pm 2e-2$ to $\mathbf{0.130} \pm 2e-3$. On the MULTI benchmark, we reach significantly better average $\mathcal{W}_1$ distances than the directly comparable literature[3], beating the previous state-of-the-art by **-15%** with a high statistical significance. This demonstrates the applicability of MMtSBM on pure cellular trajectory inference, despite the absence of restrictive modeling such as spline-valued trajectories, explicitly precomputed OT plan, or start *and* end true points trajectory pinning.

## 4.5 MMTSBM RECOVERS CONTINUOUS VIDEO DYNAMICS FROM UNPAIRED DATA

We now evaluate our method on image-space datasets, where the goal is to recover continuous trajectories (*ie videos*) from completely unpaired temporal snapshots.

### 4.5.1 MNIST DIGIT MORPHING EXPERIMENT

We conducted experiments on the MNIST dataset of hand-written digits, transporting digits in decreasing order: $4 \rightarrow 3 \rightarrow 2 \rightarrow 1 \rightarrow 0$. The algorithm was trained directly in image space, in dimension $28 \times 28 = 784$. As shown in Figure 3, MMtSBM exhibits clear digit morphing, sometimes reusing pixel structures (e.g., the top of the 3 to form the top of the 2), which is what is expected from OT in pixel space. This experiment thus demonstrates that MMtSBM manages to learn a complex temporal OT map in image space directly.

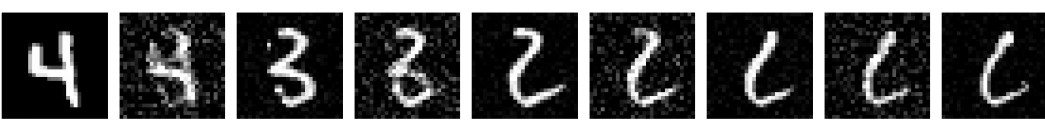

Figure 3: Video generated by MMtSBM on MNIST, backward direction. Starting image is from the test set. From left to right: generation at time $t = 4, 3.5, 3, 2.5, 2, 1.5, 1, 0.5, 0$. Integer times are marginal times.

### 4.5.2 BIOTINE CELL CULTURE EXPERIMENT

The (in-house) "biotine" dataset consists of 3-channel fluorescence images (GFP, membrane, nucleus) of A549 lung epithelial cells cultured in 384-well plates, treated with biotin, and imaged at 7 discrete time steps.

Figure 4 shows the unpaired dynamic we have at hand. We can clearly observe fluorescence loss in the cytoplasmic area, corresponding to the green channel. Interestingly, contrary to the above MNIST experiment, a mostly static *positional* evolution is observed here.

---

[3]Comparable literature: mainly methods computing the Schrödinger Bridge –but also methods performing trajectory *inference*, instead of interpolation.

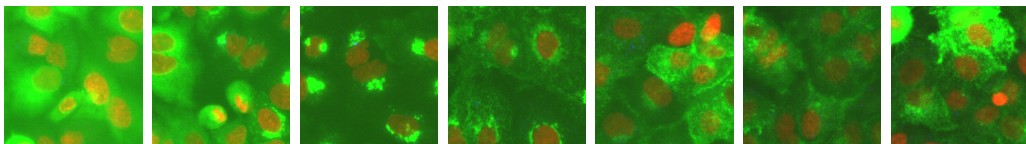

Figure 4: Ground truth biotine examples at training marginal times $t = 0, 1, 2, 3, 4, 5, 6$, from left to right.

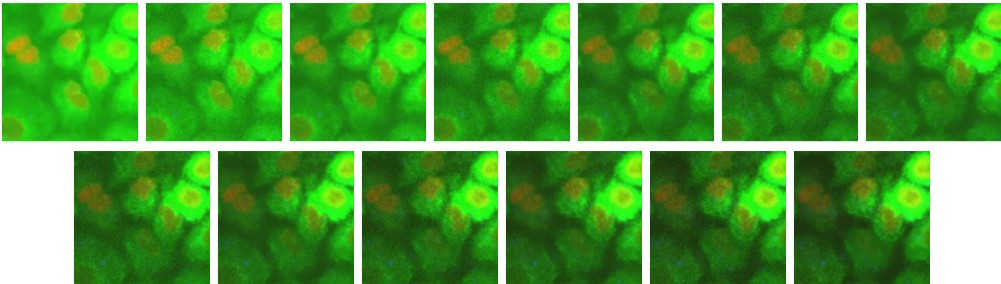

Figure 5: Video generated by MMtSBM on biotine, forward direction. To be read in reading order: top left $\rightarrow$ top right, then bottom left $\rightarrow$ bottom right. Generations at times $t = 0, 0.5, 1, 1.5, ..., 5.5, 6$. Top-left starting image is from the test set.

This actually stems from the fact that cell position is not a statistically varying information on the biotine dataset, and this (non-existing) signal is thus simply not seen by our purely unpaired method, resulting in non-moving cells. MMtSBM rather reconstructs the OT trajectory in pixel space, yielding very close cellular position while still accurately matching the time-varying phenotype (mainly: the fluorescence loss in the cytoplasm). We report in Table 4 the KID (Bińkowski et al., 2021) values of true vs generated samples, obtained with DINOv2 (Oquab et al., 2024) as a feature extractor as a baseline reference for future works.

| Time | KID ($\downarrow$) |
|---|---|
| $t = 0$ | $11.1_{\pm 0.23}$ |
| $t = 1$ | $13.0_{\pm 0.20}$ |
| $t = 2$ | $20.1_{\pm 0.23}$ |
| $t = 3$ | $23.5_{\pm 0.25}$ |
| $t = 4$ | $26.0_{\pm 0.29}$ |
| $t = 5$ | $27.7_{\pm 0.31}$ |
| all times | $17.1_{\pm 0.32}$ |

Table 4: KIDs for each marginal and all marginals together, using dinov2-vit-b-14.

To the best of our knowledge, this is the first demonstration of any method performing video generation from purely unpaired data. Together, this provides evidence for both the scalability to very high-dimensional data and for the fidelity to the underlying biological process of MMtSBM.

## 5 DISCUSSION

In this work we introduce MMtSBM, a novel method that solves the multi-marginal temporal Schrödinger Bridge problem, adapting Bridge Matching (Shi et al., 2023) to our setting. We demonstrate the theoretical soundness of both our modeling and algorithm. We show that MMtSBM indeed produces transport maps that are close to the true OT plan in toy experiments and verify its correct behavior in low-dim experiments. We achieve state-of-the-art results in 2 widely reported single-cell transcriptomic benchmarks, and for the first time demonstrate a method producing temporarily coherent videos from purely unpaired data, hoping to lead to many future scientific applications.

In future works we would like to investigate other regularizations, such as lifting the process to acceleration space to obtain smoother interpolation trajectories, or exploring other empirical reference processes than the Brownian motion. We also intend to investigate learning the transport map in a latent space. We would also like to explore using the single network theory developed in Bortoli et al. (2024) for efficiency gains, as well as simulation-free methods.

## LLM Usage Disclosure

LLMs have been used in this work for translation and redaction help, for web search of relevant references and existing literature, and for some annex coding tasks like help on visualizations.

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

CONTENTS

# A APPENDIX

## A.1 ADDITIONAL BACKGROUND

**Reciprocal projection.** The reciprocal class $\mathcal{R}(\mathbb{Q})$ consists of mixtures of $\mathbb{Q}$-bridges. For $\mathbb{P} \in \mathcal{P}(C)$,

$$\text{proj}_{\mathcal{R}(\mathbb{Q})}(\mathbb{P}) = \mathbb{P}_{0,T}\, \mathbb{Q}_{|0,T}.$$

**Markovian projection.** The Markov class $\mathcal{M}$ consists of diffusions $dX_t = v(t, X_t)\, dt + \sigma\, dB_t$. The projection $\text{proj}_{\mathcal{M}}(\Pi)$ has drift

$$dX_t = \left[ \frac{\mathbb{E}_\Pi[X_T \mid X_t] - X_t}{T - t} \right] dt + \sigma\, dB_t.$$

**Variational formulations.** Both projections solve KL problems:

$$\text{proj}_{\mathcal{R}(\mathbb{Q})}(\mathbb{P}) = \underset{\Pi \in \mathcal{R}(\mathbb{Q})}{\arg\min}\, \text{KL}(\mathbb{P} \,\|\, \Pi), \qquad \text{proj}_{\mathcal{M}}(\Pi) = \underset{M \in \mathcal{M}}{\arg\min}\, \text{KL}(\Pi \,\|\, M).$$

**Bridge matching.** In practice, the Markov drift is learned by minimising

$$\mathcal{L}(\theta) = \int_0^T \mathbb{E}_{(X_0, X_T) \sim \Pi_{0,T},\, X_t \sim \mathbb{Q}(\cdot | X_0, X_T)} \left[ \| v_\theta(X_t, t) - \tfrac{X_T - X_t}{T - t} \|^2 \right] dt.$$

**Iterative Proportional Fitting (IPF).** IPF alternately enforces marginals by KL minimisation:

$$\mathbb{P}^{2n+1} = \underset{\mathbb{P}:\mathbb{P}_T = \mu_T}{\arg\min}\, \text{KL}(\mathbb{P} \,\|\, \mathbb{P}^{2n}), \quad \mathbb{P}^{2n+2} = \underset{\mathbb{P}:\mathbb{P}_0 = \mu_0}{\arg\min}\, \text{KL}(\mathbb{P} \,\|\, \mathbb{P}^{2n+1}).$$

Unlike IMF, this requires caching full trajectories.

## A.2 OTHER PROPERTIES ON IMFF OR MMSB

**Proposition A.1** (Markov implies reciprocal). *Any Markov measure on $C([0, T], \mathbb{R}^d)$ is reciprocal. Hence $P^\star \in \mathcal{R}(Q)$. See Proposition 2.3 in Léonard (2012).*

**Proposition A.2** (Sampling with ODE probability flow). *Given the forward and backward drifts of the multi-marginal Schrödinger bridge, one can simulate trajectories using the probability flow ODE ((Song et al., 2021)):*

$$\frac{dX_t}{dt} = f_t(X_t) - \tfrac{1}{2}\sigma_t^2 \nabla \log p_t(X_t).$$

*Although the score function $\nabla \log p_t$ is not directly available, (De Bortoli et al., 2021) show that it can be equivalently recovered by averaging the forward and backward drifts:*

$$v_t(x) \;=\; \tfrac{1}{2}\Big( v_t^{\text{fwd}}(x) + v_t^{\text{bwd}}(x) \Big) \tag{10}$$

*Simulating the ODE with drift $v_t$ thus yields a deterministic sampling procedure that preserves the marginals of the stochastic bridge, providing an efficient and numerically stable alternative to direct SDE simulation.*

## A.3 CONCRETE ALGORITHMS

We always start trainings with a warmup phase, akin to Shi et al. (2023). It allows MMtSBM to start rectifying the trajectories from a non-random state, which could be complicated because the IMFF phase uses the forward/backward network to train the backward/forward one.

---

**Algorithm 2** Warmup (our algorithm)

---

1: **Input:** Subdivision $\{0 = t_0 < t_1 < \cdots < t_n = T\}$, datasets $\{\pi_{t_i}\}$, networks $v_\theta, v_\phi$, initial params $\theta, \phi$, batch size $B$, warmup steps $N_{\text{warmup}}$
2: Define bridges $\mathcal{B} = \{(t_i, t_{i+1})\}$, $b \leftarrow B/|\mathcal{B}|$
3: **for** direction $\in \{\text{forward}, \text{backward}\}$ **do**
4:      **for** $n \in [\![0, N_{\text{warmup}}]\!]$ **do**
5:          **for all** $(t_i, t_{i+1}) \in \mathcal{B}$ **in parallel do**
6:              Sample $(X_{t_i}, X_{t_{i+1}}) \sim (\pi_{t_i} \otimes \pi_{t_{i+1}})^{\otimes b}$, $t_{(i)} \sim \text{Unif}[t_i, t_{i+1}]^{\otimes b}$
7:          **end for**
8:          Aggregate $X_{\text{init}}, X_{\text{final}}, t$; Sample $Z \sim \mathcal{N}(0, I)^{\otimes B}$
9:          $X_t \leftarrow \text{Interp}_t(X_{\text{init}}, X_{\text{final}}, Z)$          ▷ cf. equation 11
10:          Update $\theta$ if forward with $\ell^{\text{fwd}}$ equation 13, else $\phi$ with $\ell^{\text{bwd}}$ equation 14
11:      **end for**
12: **end for**
13: **Output:** Warmup parameters $\theta, \phi$

---

**Algorithm 3** Iterative Markovian Factorized Fitting (IMFF) (our algorithm)

---

1: **Input:** Subdivision $\{0 = t_0 < t_1 < \cdots < t_n = T\}$, datasets $\{\pi_{t_i}\}$, networks $v_\theta, v_\phi$, warmup params $\theta, \phi$, batch size $B$, finetune steps $N_{\text{finetune}}$, inner steps $N_{\text{inner}}$
2: Define bridges $\mathcal{B} = \{(t_i, t_{i+1})\}$, $b \leftarrow B/|\mathcal{B}|$
3: **for** $N \in [\![0, N_{\text{finetune}}]\!]$ **do**
4:      **for all** $(t_i, t_{i+1}) \in \mathcal{B}$ **in parallel do**
5:          Sample $(X_{t_i}, X_{t_{i+1}})$ from $(\pi_{t_i} \otimes \pi_{t_{i+1}})^{\otimes b}$
6:          Sample $t_{(i)} \sim \text{Unif}[t_i, t_{i+1}]^{\otimes b}$
7:      **end for**
8:      Aggregate $X_{\text{init}}, X_{\text{final}}, t$
9:      **for** direction $\in \{\text{backward}, \text{forward}\}$ **do**
10:          **for** $n \in [\![0, N_{\text{inner}}]\!]$ **do**
11:              **if** direction = forward **then**
12:                  $\hat{X}_{\text{init}} \leftarrow \text{SDE}(X_{\text{final}}, v_\phi)$          ▷ cf. equation 12
13:                  $X_t \leftarrow \text{Interp}_t(\hat{X}_{\text{init}}, X_{\text{final}}, Z)$          ▷ cf. equation 11
14:                  Update $\theta$ with $\ell^{\text{fwd}}$ equation 13
15:              **else**
16:                  $\hat{X}_{\text{final}} \leftarrow \text{SDE}(X_{\text{init}}, v_\theta)$          ▷ cf. equation 12
17:                  $X_t \leftarrow \text{Interp}_t(X_{\text{init}}, \hat{X}_{\text{final}}, Z)$          ▷ cf. equation 11
18:                  Update $\phi$ with $\ell^{\text{bwd}}$ equation 14
19:              **end if**
20:          **end for**
21:      **end for**
22: **end for**
23: **Output:** Finetuned parameters $\theta, \phi$

---

### A.4 CRITICAL IMPLEMENTATION CONSIDERATIONS

A naive implementation of the algorithm quickly led to the *forgetting* of paths between marginals as training progressed. To overcome this, we developed a fully vectorized implementation that ensures stable learning across all intervals. This design is essential for the quality of our solution. Key components are detailed below.

#### A.4.1 SCALABILITY WITH HIGH DIMENSIONS AND MANY MARGINALS

Both Markovian and reciprocal projections are implemented in a fully vectorized manner. Instead of looping over intervals, all pairs are aggregated into global vectors and processed simultaneously on GPU.

At iteration $n$, for interval $[t_i, t_{i+1}]$, pairs are sampled as

$$z_i \sim (M^n)_{t_i}, \quad z_{i+1} \sim \mu_{i+1} \quad \text{(forward)}, \qquad z_{i+1} \sim (M^n)_{t_{i+1}}, \quad z_i \sim \mu_i \quad \text{(backward)}.$$

Pairs from all intervals form two batched vectors $(Z_{\text{init}}, Z_{\text{final}})$. Each bridge is then simulated in parallel as

$$X_t^{(b)} = (1 - s)\, z_{\text{init}}^{(b)} + s\, z_{\text{final}}^{(b)} + \sigma_t \sqrt{s(1-s)}\, \xi^{(b)}, \qquad \xi^{(b)} \sim \mathcal{N}(0, I).$$

This parallelization makes multi-marginal training feasible at scale.

#### A.4.2 MASKING AND TIME DISCRETIZATION

The horizon $[0, T]$ is discretized into $N_{\text{total}}$ steps, allocated proportionally to interval length:

$$N_i = \left\lfloor N_{\text{total}} \frac{t_{i+1} - t_i}{T} \right\rfloor, \quad dt_i = \pm \frac{\Delta \tau}{t_{i+1} - t_i}, \ \Delta \tau = \frac{T_{\max} - T_{\min}}{N_{\text{total}}}.$$

This ensures consistent integration with bounded cost.

Since $N_i$ varies across intervals, all trajectories are embedded into a common tensor of shape `(num_bridges, max_N)` with binary masks:

$$z_{k+1}^{(b)} = z_k^{(b)} + v(z_k^{(b)}, t_k^{(b)})\, dt^{(b)} + \sigma_{t_k^{(b)}} \sqrt{dt^{(b)}}\, \xi^{(b)},$$

updated only where `mask=1`. This allows heterogeneous bridges to evolve in a single GPU loop.

#### A.4.3 INTERPOLATION OPERATOR AND LOSSES

For each bridge $(t_i, t_{i+1})$ and batch $B$, define

$$\mathbf{s} = \frac{\mathbf{t} - t_{\text{init}}}{t_{\text{final}} - t_{\text{init}}} \in [0, 1]^B.$$

Then the interpolation is

$$\text{Interp}_{\mathbf{t}}(X_{\text{init}}, X_{\text{final}}, Z) = (1 - \mathbf{s}) \odot X_{\text{init}} + \mathbf{s} \odot X_{\text{final}} + \sqrt{\varepsilon (1 - \mathbf{s}) \odot \mathbf{s}} \odot Z, \qquad (11)$$

with $\odot$ the elementwise product.

We also define a generic simulation operator for SDEs. Given an initial condition $X_{\text{init}}$ and a drift $v_{\text{direction}}$ (either forward or backward), we denote

$$\text{SDE}(X_{\text{init}}, v_{\text{direction}}) : \quad dX_t = v_{\text{direction}}(t, X_t)\, dt + \sigma_t\, dB_t, \qquad X_{t_{\text{init}}} = X_{\text{init}}. \qquad (12)$$

This operator returns a trajectory $(X_t)_{t \in [t_{\text{init}}, t_{\text{final}}]}$.

Forward/backward losses enforce vectorized drift consistency:

$$\ell^{\text{fwd}}(\theta; \mathbf{t}, X_{\text{final}}, X_t) = \frac{1}{B} \left\| v_\theta(\mathbf{t}, X_t) - \frac{X_{\text{final}} - X_t}{t_{\text{final}} - \mathbf{t}} \right\|^2 \qquad (13)$$

$$\ell^{\text{bwd}}(\phi; \mathbf{t}, X_{\text{init}}, X_t) = \frac{1}{B} \left\| v_\phi(\mathbf{t}, X_t) - \frac{X_{\text{init}} - X_t}{\mathbf{t} - t_{\text{init}}} \right\|^2 \qquad (14)$$

#### A.4.4 TIME-DEPENDENT DRIFT NETWORKS

The drifts $v_\theta, v_\phi$ are parameterized by networks with explicit time encodings (sinusoidal, Gaussian Fourier, FiLM). This enables (i) generalization across intervals through parallel training, and (ii) sensitivity to local temporal position, ensuring bridge consistency and global coherence.

## A.5 EXPERIMENTS DETAILS

The Adam (Kingma & Ba, 2017) or AdamW (Loshchilov & Hutter, 2019) optimizer is used with a learning rate of $2 * 10^{-4}$, and SiLU activations are applied on each layers unless stated otherwise.

### A.5.1 EXACT OT BETWEEN GAUSSIAN MIXTURES

In Figure 6 we can see the (*exact*) "glued" OT plan empirically computed with POT. Observe how the global trajectory transports each Gaussian component of the mixture to a single other Gaussian component of the next marginal, yielding paths without any crossing. Note that the *true multi-marginal* transport plan remains unknown even in this simple Gaussian mixture setting.

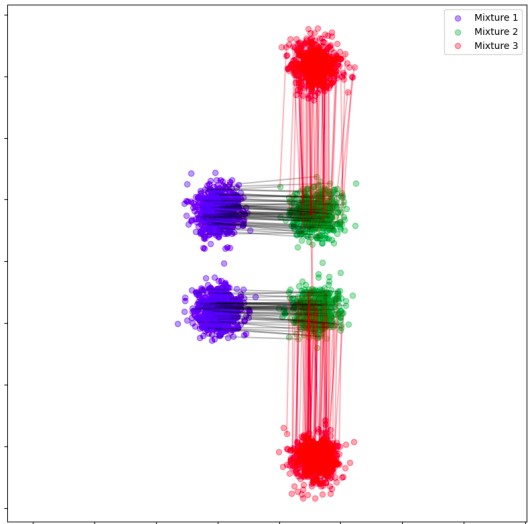

Figure 6: Here we computed the OT plan between each pair of adjacent marginals empirically, in red and black lines. This plan can serve as a good proxy for the true multi-marginal plan.

### A.5.2 8GAUSSIANS AND MOONS EXPERIMENT

We used the same experimental setting as (Shi et al., 2023), except that we increase the batch size proportionally to the number of intermediate bridges. The 2-Wasserstein distance are computed with `pot` and the integrated path energy are computed with $\mathbb{E}\left[\int_0^T \|v(t, \mathbf{Z}_t)\|^2 \, dt\right]$ where $Z_t$ is the process simulated along the ODE drift 10.

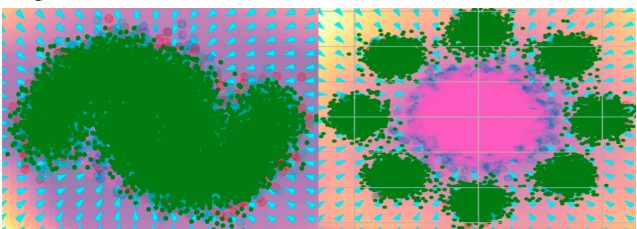

Figure 7: Third marginal fitting for the moons and 8-Gaussian trajectories. Blue vectors indicate the drift direction, with gradient intensity showing vector field strength; green points denote moving samples, and pink highlights the Gaussian fitted along the trajectory.

### A.5.3 50D GAUSSIAN EXPERIMENTS

On an NVIDIA A100 GPU, the full training took approximately 300 minutes for 30 outer iterations, each with 10,000 training steps and 20 diffusion steps per bridge.

### A.5.4   100D TRANSCRIPTOMIC EXPERIMENTS: EMBRYOID BODY

The dataset comprises 5 timepoints (Day 0 to Day 24) covering the progression from a homogeneous stem-cell population toward mesoderm, endoderm, and ectoderm precursors. The Embryoid Body dataset thus constitutes a realistic and challenging testbed for Schrödinger bridge methods, combining high dimensionality, non-Gaussian distributions, and branching lineages. We preprocessed the data following (Tong et al., 2020).

All datasets were standardized (zero mean and unit variance), and from each dataset 1000 samples were withheld to form a test set used for evaluating the Maximum Mean Discrepancy (MMD) and the Sliced Wasserstein Distance (SWD) between test set and generated samples.

We trained a network of about $300k$ parameters for 20 outer iterations with 20,000 inner iterations.

We show in Table 5 the performance advantage of our method compared to an iterative algorithm such as Chen et al. (2023a).

|  | DSBM (Chen et al., 2023a) | | MMtSBM (ours) | |
|---|---|---|---|---|
| number of marginals | 4 | 5 | 4 | 5 |
| Train time | 33min | 44 min | 20 min | 32 min |
| Sampling time | 2.00s | 2.02s | 2.00 s | 2.00 s |

Table 5: Training and sampling times for Chen et al. (2023a) and MMtSBM (ours) in dimension 100.

### A.5.5   100D TRANSCRIPTOMIC EXPERIMENTS: MULTI

We reused the preprocessed data from Tong et al. (2024b). We do not whiten it. We conducted a minimal sweep to select the best $\sigma$ (0.3). The network is a simple 3-layers MLP with around $500k$ parameters and we employ 150 discretization time steps in total. Metrics are computed over $1k$ true test samples vs $1k$ generated samples, where these generations themselves come from the previous test marginal. We trained 3 models with different seeds for each left-out time (either $t = 1$ or $t = 2$, corresponding to days 3 and 4). Our reported standard deviation is the pooled variance of the best same-hyperparameters $\sigma = 0.3$ models over 2 groups, each group corresponding to a left-out time. Other papers seem to have reported the overall variance, which we think makes less sense given the structure of the problem.

| Group | Number of runs | Mean | Std |
|---|---|---|---|
| Leave-out & test $t = 1$ | 3 | 37.026 | 0.822 |
| Leave-out & test $t = 2$ | 3 | 52.059 | 0.367 |
| Global | 6 | 44.542 | 0.637 |

Table 6: Per-group statistics with pooled standard deviation $s_{\text{pooled}} = \sqrt{\sum (n_i - 1)s_i^2 \, / \, \sum (n_i - 1)}$, where $n_i$ and $s_i$ are the sample size and standard deviation of each group.
Group

About other methods reported in Table 3: only I-CFM, I-MFM$_{\text{RBF}}$, and MMtSBM (ours) do not rely on a precomputed OT plan, be it exact or approximate. GAGA (Sun et al., 2025) performs *interpolation between 2 true pinned endpoints* in the latent space of a metric-aware autoencoder trained with the true exact OT plan; we thus still claim SOTA, either within methods solving the SB, or within methods doing "pure" trajectory *inference* (without a pinned true endpoint).

### A.5.6   MNIST DIGIT MORPHING EXPERIMENT

We experimented 2 approaches for the MNIST dataset: a MLP with flattened image vectors of dimension ($28 \times 28 = 784$, and a UNet with image-shape data of shape $(28, 28)$.

### A.5.7   BIOTINE CELL CULTURE EXPERIMENT

We perform learning directly in image space at $3 \times 128 \times 128$ definition with a 3M parameters UNet. We also experimented with learning in a VAE latent space but produced images were more blurry.

| Dataset | Biotine |
|---|---|
| Dimension | $128 \times 128 \times 3 = 49,152$ |
| Number of marginals | 6 |
| Training time | 5 h |
| Number of epochs | 5 |
| Sampling time | 32 s |
| Generated frames | 602 |

Table 7: Training and sampling statistics for video generation on the Biotine dataset.

The model trains in only 5 hours and subsequently generates an entire 602-frame trajectory in just 32 seconds, demonstrating both low training cost and highly efficient sampling.

## A.6 PROOFS

### A.6.1 DEFINITION 3.2

*Proof of variational proposition in Definition 3.2 (variational characterization).* By the additive property of the KL divergence (Léonard, 2014), for any $P \in \mathcal{P}(C([0,T], \mathbb{R}^d))$ and $\Pi \in \mathcal{R}^{\otimes}(\mathbb{Q})$, we can write

$$KL(P \,\|\, \Pi) = KL(P_{t_0,\dots,t_K} \,\|\, \Pi_{t_0,\dots,t_K}) + \mathbb{E}_{P_{t_0,\dots,t_K}} \left[ KL\Big( P_{[0,T]}^{x_0,\dots,x_K} \,\|\, \bigotimes_{i=0}^{K-1} \mathbb{Q}_{[t_i,t_{i+1}]}^{x_i,x_{i+1}} \Big) \right],$$

where $P_{[0,T]}^{x_0,\dots,x_K}$ denotes the conditional law of $P$ given its values at the grid points $(t_0,\dots,t_K)$.

Restricting to $\Pi$ such that $\Pi_{t_0,\dots,t_K} = P_{t_0,\dots,t_K}$ cancels the first KL term, and then the minimizer is uniquely obtained by replacing the conditional path law of $P$ with the tensor product of $Q$-bridges between each $(x_i, x_{i+1})$.

Hence the optimal projection is

$$\Pi^{\star} = P_{t_0,\dots,t_K} \bigotimes_{i=0}^{K-1} \mathbb{Q}_{[t_i,t_{i+1}]}^{x_i,x_{i+1}},$$

which is exactly the definition of the factorized reciprocal projection. $\qquad\square$

### A.6.2 DEFINITION 3.3

*Proof of proposition in the Definition 3.3 in the Brownian case.* By Definition 3.3, the effective drift is

$$v_t^{\star}(x) = \sigma_t^2 \, \mathbb{E}_{\Pi_{t_{i+1}|t}} \left[ \nabla \log Q_t^{|t_i,t_{i+1}}(X_{t_{i+1}} \mid X_t) \,\Big|\, X_t = x \right].$$

For a Brownian reference process, the transition kernel is Gaussian,

$$Q_t^{|t_i,t_{i+1}}(y \mid x) = \frac{1}{(2\pi\sigma^2(t_{i+1} - t))^{d/2}} \exp\Big( -\frac{\|y - x\|^2}{2\sigma^2(t_{i+1} - t)} \Big),$$

so that

$$\nabla_x \log Q_t^{|t_i,t_{i+1}}(y \mid x) = \frac{y - x}{\sigma^2(t_{i+1} - t)}.$$

Plugging this into the definition yields

$$v_t^{\star}(x) = \sigma_t^2 \, \mathbb{E}\left[ \frac{X_{t_{i+1}} - x}{\sigma^2(t_{i+1} - t)} \,\Big|\, X_t = x \right].$$

In the Brownian case $\sigma_t^2 = \sigma^2$, which simplifies to

$$v_t^{\star}(x) = \frac{\mathbb{E}[X_{t_{i+1}} \mid X_t = x] - x}{t_{i+1} - t},$$

as claimed. $\qquad\square$

### A.6.3 PROPOSITION 3.1

*Proof of Proposition 3.1.* The feasible set

$$\mathcal{A} = \{P : P \ll Q,\ P_{t_i} = \mu_{t_i},\ i = 0, \ldots, K\}$$

is convex. Since the functional $P \mapsto D_{\mathrm{KL}}(P\|Q)$ is strictly convex, there is at most one minimizer.

To show existence, observe that $\mathcal{A}$ is non-empty. Indeed, consider any coupling $\gamma$ of $(\mu_{t_0}, \ldots, \mu_{t_K})$. For each pair $(x_i, x_{i+1})$, let $Q_{[t_i, t_{i+1}]}^{x_i, x_{i+1}}$ denote the Brownian bridge of $Q$ conditioned on $X_{t_i} = x_i$ and $X_{t_{i+1}} = x_{i+1}$. Then the measure

$$P = \int \bigotimes_{i=0}^{K-1} Q_{[t_i, t_{i+1}]}^{x_i, x_{i+1}}\, d\gamma(x_0, \ldots, x_K)$$

belongs to $\mathcal{A}$. Hence the admissible set is non-empty.

Therefore, (MMSB) admits a unique solution $P^\star$. □

### A.6.4 PROPOSITION 3.2

*Proof of Proposition 3.2.* The argument is identical to Proposition 2.10 in Léonard (2014), extended to the multi-marginal setting. For any admissible path measure $P \ll Q$, the additivity property of the relative entropy gives

$$KL(P \,\|\, Q) = KL(P_{t_0, \ldots, t_K} \,\|\, Q_{t_0, \ldots, t_K}) + \mathbb{E}_{P_{t_0, \ldots, t_K}}[KL(P(\cdot \mid X_{t_0}, \ldots, X_{t_K}) \,\|\, Q(\cdot \mid X_{t_0}, \ldots, X_{t_K}))].$$

Since the second term is always nonnegative, minimizing the dynamic problem is equivalent to minimizing the static one. Moreover, the inequality becomes an equality if and only if

$$P(\cdot \mid X_{t_0}, \ldots, X_{t_K}) = Q(\cdot \mid X_{t_0}, \ldots, X_{t_K}), \quad P_{t_0, \ldots, t_K}\text{-a.s.}$$

Hence the optimal dynamic solution $P^\star$ is uniquely obtained from the optimal static solution $\pi^\star$ by gluing the conditional bridges of $Q$, which establishes the equivalence. □

### A.6.5 PROPOSITION 3.3

*Proof of Proposition 3.3.* We follow the argument of (Léonard, 2014, Prop. 2.10). Fix an intermediate time $t_k$ with $0 < k < n$. For any $Q \in \mathcal{P}(\Omega)$ and $z \in X$, set

$$Q_{[0, t_k]}^{t_k, z} := Q(X_{[0, t_k]} \in \cdot \mid X_{t_k} = z), \qquad Q_{[t_k, 1]}^{t_k, z} := Q(X_{[t_k, 1]} \in \cdot \mid X_{t_k} = z).$$

Let $\mu \in \mathcal{P}(X)$ and for each $z \in X$ prescribe $Q_z^< \in \mathcal{P}(\Omega_{[0, t_k]} \cap \{X_{t_k} = z\})$, $Q_z^> \in \mathcal{P}(\Omega_{[t_k, 1]} \cap \{X_{t_k} = z\})$. By the entropy additivity property (see formula (A.8) in Léonard (2014)), the measure

$$P^* = \int_X Q_z^< \otimes Q_z^>\, \mu(dz)$$

is the unique minimizer of $H(\cdot \mid R)$ under these constraints, and it satisfies

$$P_{[t_k, 1]}^*(\cdot \mid X_{[0, t_k]}) = P_{[t_k, 1]}^*(\cdot \mid X_{t_k}).$$

This is exactly the Markov property at time $t_k$.

Now apply this to $Q = \widehat{P}$, the solution of the multi-marginal Schrödinger problem. If $\widehat{P}$ were not Markov, one could construct a measure $P^*$ with the same time-marginal constraints but strictly smaller entropy, a contradiction with the definition of a minimizer. Since $t_k$ was arbitrary, $\widehat{P}$ must be Markov at all grid times $t_0, \ldots, t_n$, hence Markov on $[0, 1]$. □

### A.6.6   PROPOSITION 3.4

*Proof of Proposition 3.4.* The argument is a direct extension of Theorem 2.8 and Proposition 2.10 in Léonard (2014).

Assume that the reference law $Q_{t_0,\dots,t_K}$ satisfies the usual regularity conditions: (i) each one-time marginal coincides with a reference measure $m$; (ii) there exists a nonnegative function $A$ such that

$$Q_{t_0,\dots,t_K}(dx_0,\dots,dx_K) \geq \exp\Big(-\sum_{i=0}^{K} A(x_i)\Big)\, m(dx_0)\cdots m(dx_K);$$

(iii) there exists $B$ such that

$$\int_{\mathcal{X}^{K+1}} \exp\Big(-\sum_{i=0}^{K} B(x_i)\Big) Q_{t_0,\dots,t_K}(dx_0,\dots,dx_K) < \infty;$$

(iv) either $m^{\otimes(K+1)} \ll Q_{t_0,\dots,t_K}$ or the converse holds. Suppose further that the prescribed marginals $(\pi_{t_0},\dots,\pi_{t_K})$ satisfy $H(\pi_{t_i}\,|\,m) < \infty$,

$$\sum_{i=0}^{K} \int (A+B)(x)\, d\pi_{t_i}(x) < \infty,$$

and that they are internal in the sense of Proposition 2.6 of (Léonard, 2014).

Under these assumptions, the dual problem is well posed. Introducing Lagrange multipliers $(\varphi_i)_{i=0}^{K}$ for the marginal constraints, convex duality shows that the minimizer $\pi^\star$ of the static problem is absolutely continuous with respect to $Q_{t_0,\dots,t_K}$ with density

$$\frac{d\pi^\star}{dQ_{t_0,\dots,t_K}}(x_0,\dots,x_K) = \exp\Big(\sum_{i=0}^{K} \varphi_i(x_i)\Big).$$

Defining $f_i(x_i) := e^{\varphi_i(x_i)}$ yields the factorized form

$$\frac{d\pi^\star}{dQ_{t_0,\dots,t_K}}(x_0,\dots,x_K) = \prod_{i=0}^{K} f_i(x_i).$$

$\square$

### A.6.7   PROPOSITION 3.6

*Proof of Proposition 3.6.* The argument is the same as in the two-marginal case (Shi et al., 2023, Prop. 2), except that all computations must now be performed interval by interval along the grid $t_0 < \cdots < t_K$. Under Assumptions A1–A3, the Doob–$h$ transform is well-defined on each interval $[t_i, t_{i+1}]$ and Lemma 11 of Shi et al. (2023) applies verbatim. The only change is that the terminal conditioning in the backward equation is at $t_{i+1}$ instead of $T$. This yields the drift

$$v_t^\Pi(x) = \sigma_t^2\, \mathbb{E}_\Pi\Big[\nabla \log Q_{t_{i+1}|t}^{t_i,t_{i+1}}(X_{t_{i+1}}\mid X_t)\,\big|\, X_{t_i}, X_t\Big], \qquad t \in [t_i, t_{i+1}].$$

Hence the dynamics of $\Pi$ is piecewise independent: its increment on $[t_i, t_{i+1}]$ depends only on the local bridge $Q^{t_i,t_{i+1}}$.

The same interval-wise independence holds for any Markov $M \in \mathcal{M}$, whose SDE also factorizes on the grid. Thus both $\Pi$ and $M$ have product decompositions over the intervals, and their Radon–Nikodym derivative factorizes multiplicatively,

$$\frac{d\Pi}{dM} = \prod_{i=0}^{K-1} \frac{d\Pi^{(i)}}{dM^{(i)}}.$$

Taking logarithms and integrating with respect to $\Pi$ gives the additivity of the relative entropy,

$$KL(\Pi\|M) = \sum_{i=0}^{K-1} KL(\Pi^{(i)}\|M^{(i)}).$$

For each interval $[t_i, t_{i+1}]$, using the conditional expectation identity as in the proof of Shi et al. (2023), we have for every $t \in [t_i, t_{i+1}]$,

$$\mathbb{E}_{\Pi_{t_i,t}} \left[ \left\| \sigma_t^2 \, \mathbb{E}_{\Pi_{t_{i+1}|t_i,t}} [\nabla \log Q_{t_{i+1}|t}^{t_i,t_{i+1}}(X_{t_{i+1}} \mid X_t) \mid X_t, X_{t_i}] - v_t(X_t) \right\|^2 \right]$$

$$\geq \mathbb{E}_{\Pi_{t_i,t}} \left[ \left\| \sigma_t^2 \mathbb{E}_{\Pi_{t_{i+1}|t}} [\nabla \log Q_{t_{i+1}|t}^{t_i,t_{i+1}}(X_{t_{i+1}} \mid X_t) \mid X_t, X_{t_i}] - v_t^\star(X_t) \right\|^2 \right],$$

where the optimal drift is defined by the orthogonal projection

$$v_t^\star(x) = \sigma_t^2 \, \mathbb{E}_{\Pi_{t_{i+1}|t}} \left[ \nabla \log Q_{t_{i+1}|t}^{t_i,t_{i+1}}(X_{t_{i+1}} \mid X_t) \mid X_t = x_t \right].$$

Using Léonard (2012), Theorem 2.3 on each interval and summing the contributions gives

$$KL(\Pi \| M^\star) = \frac{1}{2} \sum_{i=0}^{K-1} \int_{t_i}^{t_{i+1}} \mathbb{E}_{\Pi_t} \left[ \|v_t^\Pi(X_t) - v_t^\star(X_t)\|^2 / \sigma_t^2 \right] dt.$$

Finally, the same Fokker–Planck uniqueness argument as in Shi et al. (2023) ensures that $M_t^\star = \Pi_t$ for all $t \in [t_i, t_{i+1}]$ and all $i$. Since the grid points are included, this implies $M^\star = \Pi$, which concludes the proof. □

### A.6.8 LEMMA 3.1

*Proof of Lemma 3.1.* For the Markovian part, the equality follows analogously to the proof of (Shi et al., 2023).

For each interval $[t_i, t_{i+1}]$, the same quadratic expansion gives

$$2\,KL\left(\Pi^{(i)} \,\middle\|\, M^{(i)}\right) = 2\,KL\left(\Pi^{(i)} \,\middle\|\, \mathrm{proj}_{\mathcal{M}}(\Pi)^{(i)}\right) + 2\,KL\left(\mathrm{proj}_{\mathcal{M}}(\Pi)^{(i)} \,\middle\|\, M^{(i)}\right).$$

Summing this identity over $i = 0, \ldots, K-1$, using the interval-wise independence, yields

$$2\,KL(\Pi \| M) = 2\,KL(\Pi \| \mathrm{proj}_{\mathcal{M}}(\Pi)) + 2\,KL(\mathrm{proj}_{\mathcal{M}}(\Pi) \| M),$$

which is the desired result.

For the factorized reciprocal part :

Let $\Pi \in \mathcal{R}^\otimes(Q)$ and denote by

$$\Pi^\star = \mathrm{proj}_{\mathcal{R}^\otimes(Q)}(\mathbb{P}) = \mathbb{P}_{t_0,\ldots,t_K} \otimes_{i=0}^{K-1} Q_{[t_i,t_{i+1}]}^{x_i,x_{i+1}}.$$

We have the Radon–Nikodym factorization

$$\frac{d\mathbb{P}}{d\Pi} = \frac{d\mathbb{P}}{d\Pi^\star} \cdot \frac{d\Pi^\star}{d\Pi}(X_{t_0}, \ldots, X_{t_K}).$$

By integrating w.r.t. $\mathbb{P}$ and applying Csiszár's Pythagorean identity (Csiszár, 1975, Eq. 2.6), we obtain

$$KL(\mathbb{P}\|\Pi) = KL(\mathbb{P}\|\Pi^\star) + \int \log \frac{d\Pi^\star}{d\Pi}(x_0, \ldots, x_K) \, d\mathbb{P}_{t_0,\ldots,t_K}.$$

Since $\mathbb{P}_{t_0,\ldots,t_K} = \Pi^\star_{t_0,\ldots,t_K}$, the second term equals

$$\int \log \frac{d\Pi^\star}{d\Pi}(x_0, \ldots, x_K) \, d\Pi^\star_{t_0,\ldots,t_K} = KL(\Pi^\star\|\Pi).$$

Thus

$$KL(\mathbb{P}\|\Pi) = KL(\mathbb{P}\|\Pi^\star) + KL(\Pi^\star\|\Pi),$$

which concludes the proof. □

### A.6.9  PROPOSITION 3.7

*Proof of Proposition 3.7.* It follows from the fact that the time-reversal map $\mathcal{T} : \Omega \to \Omega$ is a bijection, and by reversibility of the reference process $\mathbb{Q}$ we have, for any probability measure $\mathbb{P} \in \mathcal{P}(C)$,

$$KL(\mathbb{P} \,\|\, \mathbb{Q}) = KL(\mathbb{P} \circ \mathcal{T} \,\|\, \mathbb{Q} \circ \mathcal{T}) = KL(\mathbb{P} \circ \mathcal{T} \,\|\, \mathbb{Q}).$$

To prove the direction "$\Longrightarrow$", assume $\mathbb{P} \in \mathcal{R}^{\otimes}(\mathbb{Q})$ is the minimizer of the forward problem. Then, for any $\Pi \in \mathcal{R}^{\otimes}(\mathbb{Q})$ we have $\Pi \circ \mathcal{T} \in \mathcal{R}^{\otimes}(\mathbb{Q})$, and

$$KL(\Pi \,\|\, \mathbb{Q}) = KL(\Pi \circ \mathcal{T} \,\|\, \mathbb{Q} \circ \mathcal{T}) \;\geq\; KL(\mathbb{P} \circ \mathcal{T} \,\|\, \mathbb{Q} \circ \mathcal{T}) = KL(\mathbb{P} \,\|\, \mathbb{Q}).$$

The reverse direction follows by symmetry, replacing $\mathbb{P}$ with $\mathbb{P} \circ \mathcal{T}$. Thus, working with forward or backward processes is equivalent up to the bijection $\mathcal{T}$, and the KL minimization problem is unchanged. In particular, this justifies that alternating forward and backward steps in the IMFF algorithm is well-defined and analogous to IPF. $\qquad\square$

### A.6.10  PROPOSITION 3.8

*Proof of Proposition 3.8, first claim.* As a reminder, we follow the same argument as in (Shi et al., 2023) and (De Bortoli et al., 2021). Applying Lemma 3.1, for any $N \in \mathbb{N}$ we obtain

$$KL(\mathbb{P}^0 \,\|\, \mathbb{P}^{\star}) = KL(\mathbb{P}^0 \,\|\, \mathbb{P}^1) + KL(\mathbb{P}^1 \,\|\, \mathbb{P}^2) + \cdots + KL(\mathbb{P}^N \,\|\, \mathbb{P}^{\star}).$$

Since each term is nonnegative, we deduce the monotonicity

$$KL(\mathbb{P}^{n+1} \,\|\, \mathbb{P}^{\star}) \;\leq\; KL(\mathbb{P}^n \,\|\, \mathbb{P}^{\star}),$$

and boundedness $KL(\mathbb{P}^n \,\|\, \mathbb{P}^{\star}) \leq KL(\mathbb{P}^0 \,\|\, \mathbb{P}^{\star}) < \infty$. This proves the first claim. $\qquad\square$

*Proof of Proposition 3.8, second claim.* We proceed by induction, adapting the argument of (De Bortoli et al., 2021, Appendix C.8).

At initialization, we choose $\mathbb{P}^0 \in \mathcal{R}^{\otimes}(\mathbb{Q})$ with $\mathbb{P}^0_{t_i} = \mu_{t_i}$ for all $i$. We also define $M^0 = \mathrm{proj}_{\mathcal{M}}(\mathbb{P}^0)$.

By construction (Algorithm 1), the IMFF sequence alternates:

$$\mathbb{P}^{2n+1} = \mathrm{proj}_{\mathcal{M}}(\mathbb{P}^{2n}), \qquad \mathbb{P}^{2n+2} = \mathrm{proj}_{\mathcal{R}^{\otimes}(\mathbb{Q})}(\mathbb{P}^{2n+1}).$$

Suppose now that $\mathbb{P}^{2n}$ satisfies the claim. By definition, $\mathbb{P}^{2n+1} \in \mathcal{M}$ and $\mathbb{P}^{2n+2} \in \mathcal{R}^{\otimes}(\mathbb{Q})$. From Lemma 3.1, we then have

$$KL(\mathbb{P}^{2n+1} \,\|\, P^{\star}) \leq KL(\mathbb{P}^{2n} \,\|\, P^{\star}), \qquad KL(\mathbb{P}^{2n+2} \,\|\, P^{\star}) \leq KL(\mathbb{P}^{2n+1} \,\|\, P^{\star}).$$

Hence, $(KL(\mathbb{P}^n \,\|\, P^{\star}))_{n \in \mathbb{N}}$ is a nonincreasing sequence bounded below by 0, and is therefore convergent. Moreover, by induction we have $\mathbb{P}^n \in \mathcal{M} \cap \mathcal{R}^{\otimes}(\mathbb{Q})$ for all $n$, so the limit must coincide with $P^{\star}$, the unique measure in this intersection with prescribed marginals.

Finally, note that in Algorithm 1 the forward and backward Markovian steps are time-reversals of each other (they follow the same law under the change of variable $t \mapsto T - t$). Therefore, alternating a backward step with a forward reciprocal projection, or a forward step with a backward reciprocal projection, is equivalent from the viewpoint of convergence analysis. All the arguments above apply symmetrically in both directions, and the resulting sequence $(\mathbb{P}^n)_{n \in \mathbb{N}}$ still converges.

We conclude that

$$\lim_{n \to \infty} KL(\mathbb{P}^n \,\|\, P^{\star}) = 0,$$

and $P^{\star}$ is indeed the weak solution produced by the IMFF algorithm, proving the second claim. $\quad\square$

### A.6.11 THEOREM 3.2

*Proof of Theorem 3.2.* As a reminder, the argument is the same as in (Shi et al., 2023) and (De Bortoli et al., 2021), but adapted to the multi-marginal setting.

By Proposition 3.8, the sequence $(\mathbb{P}^n)_{n \in \mathbb{N}}$ is bounded in KL divergence with respect to $\mathbb{P}^\star$, hence relatively compact under weak convergence. Thus, it admits a subsequence $(\mathbb{P}^{n_j})_j$ converging weakly to some limit $\mathbb{P}^\infty$. By construction, $\mathbb{P}^\infty \in \mathcal{M} \cap \mathcal{R}^{\otimes}(Q)$ and matches the marginals $(\mu_{t_i})_{i=0}^{K}$, so by uniqueness of the weak MMSB solution we must have $\mathbb{P}^\infty = \mathbb{P}^\star$.

By lower semicontinuity of KL, this implies

$$\lim_{n \to \infty} KL(\mathbb{P}^n \,\|\, \mathbb{P}^\star) = 0.$$

Finally, the inequality

$$KL(\mathbb{P}^{\mathrm{MMSB}} \,\|\, Q) \ \leq \ KL(\mathbb{P}^\star \,\|\, Q) \ \leq \ KL(\mathbb{P}^{\mathrm{pair}} \,\|\, Q)$$

is justified because $\mathbb{P}^{\mathrm{MMSB}}$ is the global minimizer (hence gives the smallest KL), while $\mathbb{P}^\star$ is the best Markovian candidate in $\mathcal{M} \cap \mathcal{R}^{\otimes}(Q)$, and therefore lies below the pairwise construction obtained by gluing local bridges. $\qquad\square$

