# OpenReview forum: "Multi-marginal temporal Schrödinger Bridge Matching for video generation from unpaired data"
_ICLR.cc/2026/Conference — Submitted to ICLR 2026_

### Official Review · Reviewer_DCoi · 2025-10-26

**Soundness:** 3
**Presentation:** 2
**Contribution:** 2
**Rating:** 2
**Confidence:** 4

**Summary:**

The authors proposed a multi-marginal extension of diffusion Schrodinger bridge matching to learn trajectories with multiple unpaired snapshots and applied the method in video generation from unpaired images.

**Strengths:**

It is an important problem how one could learn dynamics with multiple unpaired marginal observations and the authors proposed a method that is sensible.

**Weaknesses:**

**Theory**
- Although sound, the authors should check whether the results are already known in e.g., Lavenant et al. 2024.

**Experimental details**
- The write up feels a bit rushed and it is hard to follow how these experiments are done.

**Experimental results**
- In my experience all these methods can have some variance depends on initialization, the results reported in table 2 does not have error margins. E.g., from 0.03 to 0.02 might not be meaningful if the error bar is 0.01.
- Vanilla piece-wise SB should be compared.
- Video example in 4.5.2 is purely qualitative and has no baseline. I was not able to locate the data on the github repo linked in the paper.

**Notations**:
- The $f_i$'s in Prop 3.4 seems not defined in main text, they are exponential of Lagrangian multipliers $\varphi_i$ according to appendix. I would suggest define them in main text or move the proposition in appendix.

**References**
- I am a bit surprised Lavenant et al. 2024 (date back to 2021 on arxiv) is not referenced. Uniqueness results etc. were also discussed in this theory work quite extensively.
- Chen et al. 2019 (and subsequent papers) seems missing.
- Line 335-337, the authors internal notes [CITE] still remain.

Lavenant, H., Zhang, S., Kim, Y.H. and Schiebinger, G., 2024. Toward a mathematical theory of trajectory inference. The Annals of Applied Probability, 34(1A), pp.428-500. [arxiv:2102.09204](https://arxiv.org/abs/2102.09204)

Chen, Yongxin, Giovanni Conforti, Tryphon T. Georgiou, and Luigia Ripani. "Multi-marginal Schrödinger bridges." In International Conference on Geometric Science of Information, pp. 725-732. Cham: Springer International Publishing, 2019.

**Questions:**

- How does the method compare to Chen et al. 2019 and piecewise SB?

Chen, Yongxin, Giovanni Conforti, Tryphon T. Georgiou, and Luigia Ripani. "Multi-marginal Schrödinger bridges." In International Conference on Geometric Science of Information, pp. 725-732. Cham: Springer International Publishing, 2019.

---

> ### Author Response · Authors · 2025-11-24
> **Response 1 to Reviewer DCoi**
>
> We thank the reviewer for their knowledgeable review.
> In this response we answer to each weakness and question raised specifically here, additionally to a global response.
>
> > Although sound, the authors should check whether the results are already known in e.g., Lavenant et al. 2024.
>
> We would like to clarify that, when developing our theoretical results —especially the uniqueness of the multi-marginal solution— we were not aware of their paper.
>
> After checking their 2021 preprint, we noticed that it is built on the same classical references that we used (mainly Léonard 2012, 2014). Our work also starts from this standard Schrödinger Bridge literature, but we further introduce:
> - a multi-marginal reformulation of the classical SB problem under mild assumptions (in the spirit of Léonard’s “least decrease deviation” arguments)
> - a clear statement and proof that the solution is Markov whenever the reference process is Markov
>
> The work of [Lavenant et al. 2021](https://arxiv.org/abs/2102.09204) is very interesting and might cover our uniqueness proof from another angle. We would need further investigations into the detailed hypotheses and formulations used in this work to be sure that their quite complicated setting covers ours exactly, or not.
>
> We stress that beyond these theoretical properties, we derive a full, motivated algorithmic procedure to reach the multi-marginal solution, introducing the factorized reciprocal class and the IMFF algorithm which is not present in previous works. Whether such principled algorithms would be reachable from [Lavenant et al. 2021](https://arxiv.org/abs/2102.09204). is not clear at all. They do not mention any markovian or reciprocal projection mechanism, for example.
>
> Of course all the experimental parts are completely disjoint as they do not even train neural networks.
>
> > The write up feels a bit rushed and it is hard to follow how these experiments are done.
>
> Thank you for the feedback. We agree that the experimental section was condensed due to space limits. In the revised version, we will expand the description of the experimental pipeline, training setup, dataset preprocessing, evaluation metrics, and hyperparameters—and ensure the flow is easier to follow. We will also make the Appendix more visible in the main text by adding explicit pointers to the relevant sections, so that readers can easily locate the full experimental details and supplementary analyses.
>
> > In my experience all these methods can have some variance depends on initialization, the results reported in table 2 does not have error margins. E.g., from 0.03 to 0.02 might not be meaningful if the error bar is 0.01.
>
> You are completely right to ask for errors bars that we wanted to include and did not because of time.
>
> Here we provide the error bars for 10 evaluations (for us; 3 seeds for DMSB) on Table 2:
>
> | Time | DMSB MMD ↓ | DMSB SWD ↓ | MMtSBM (ours) MMD ↓ | MMtSBM (ours) SWD ↓ |
> |------|------------|------------|----------------------|-----------------------|
> | t₁   | 0.021      | 0.114      | 0.016            | 0.104             |
> | t₂   | 0.029      | 0.155      | 0.020            | 0.139             |
> | t₃   | 0.038      | 0.190      | 0.020            | 0.127             |
> | t₄   | 0.034      | 0.155      | 0.020            | 0.143             |
> | Average | 0.032 ± 3e−3 | 0.160 ± 2e−2 | 0.019 ± 4e−4 | 0.128 ± 2e−3 |
>
> As you can see **our margin is significant**.
>
> We will update the manuscript with this revised table and with the error bars at all marginals.
>
> > Vanilla piece-wise SB should be compared.
>
> This comparison would actually mean training $2n$ networks which is quite prohibitive (if not unfair!).
>
> We actually compare with this ideal case in 4.2 and report equivalent metrics to that theoretical case.
>
> > Video example in 4.5.2 is purely qualitative and has no baseline. I was not able to locate the data on the github repo linked in the paper.
>
> The biotine data is indeed private, as reported. This biological dataset is in-house and will simply not be made public by the lab that has produced it, unfortunately.
>
> As said to another reviewer, our contribution is precisely to show that it is possible, for the first time, to reconstruct meaningful dynamics in high-dimensional image settings without temporal supervision and solely from static marginal distributions. Since no existing approach can be applied to this specific unpaired scenario, standard quantitative comparisons are not feasible. Instead, we evaluate the method through qualitative biological consistency: preservation of spatial structure, correct fluorescence decay patterns, and realistic cytoplasmic dynamics, which are all observed in the generated videos.

---

> ### Author Response · Authors · 2025-11-24
> **Response 2 to Reviewer DCoi**
>
> > The f_i's in Prop 3.4 seems not defined in main text, they are exponential of Lagrangian multipliers according to appendix. I would suggest define them in main text or move the proposition in appendix.
>
> The functions $ f_i $ ​ in Proposition 3.4 indeed correspond to the exponentials of the Lagrange multipliers arising from the dual formulation, as made explicit in the appendix. We agree that their role should be clarified earlier in the main text. In the revised version, we will add a short definition of the $ f_i $'s  directly in the main text when presenting Proposition 3.4.
>
> > Chen et al. 2019 (and subsequent papers) seems missing.
>
> Although we do not cite this specific paper, we already reference Deep Momentum Multi-Marginal Schrödinger Bridge (2023), which builds directly on the same theory. We will add  [Chen et al. 2019](https://arxiv.org/abs/1902.08319) in the literature review.
>
> > Line 335-337, the authors internal notes [CITE] still remain.
>
> We apologize for this mistake which has been corrected in the revised manuscript.
>
> > How does the method compare to Chen et al. 2019 and piecewise SB?
>
> We stress than [Chen et al. 2019](https://arxiv.org/abs/1902.08319) does not conduct any experiments, let alone deep learning ones.
>
> A direct "follow-up experiment paper" is clearly [DMSB](https://arxiv.org/abs/2303.01751) against which we extensively compare against.
>
> The key difference with  [DMSB](https://arxiv.org/abs/2303.01751) is that their method solves the multi-marginal problem pairwise, interval by interval, and occasionally requires full 0→N or N→0 passes to patch these local solutions. This makes the procedure heavy, not theoretically aligned with the multi-marginal SB formulation, and unable to maintain the correct Markov/reciprocal structure except asymptotically.
>
> Our method instead treats all marginals simultaneously through a single unified forward–backward pass, which guarantees global temporal coherence at every iteration and preserves by construction the correct Markovian and reciprocal structure on each sub-interval. This unified view scales naturally to high-dimensional settings.
>
> Importantly, our approach is not a naïve extension of IMF (Shi et al.) obtained by gluing N two-marginal solutions. We fundamentally modified IMF through a parallelized update scheme: each forward or backward sweep produces one single global drift that satisfies all constraints at once, keeping the proper SB structure.
>
> Moreover, the phase-space formulation in DMSB (adding a velocity variable) behaves more like a spline-correction mechanism than a genuine multi-marginal Schrödinger bridge. It introduces an additional algorithmic integration layer whose role is to smooth or regularize trajectories—not something derived from the theoretical MMSB problem itself.
>
> Practically, our unified design allows us to scale where DMSB cannot: we can generate coherent trajectories directly in image space, whereas DMSB’s optimization becomes too heavy to scale, even with an auto-encoder. For equivalent benchmarks, our method also achieves better performance with a model roughly three times smaller.

---

### Official Review · Reviewer_wGJR · 2025-10-31

**Soundness:** 3
**Presentation:** 3
**Contribution:** 4
**Rating:** 8
**Confidence:** 4

**Summary:**

Manuscript addresses reconstruction of temporal dynamics from unpaired snapshots via a time-ordered multi-marginal Schrödinger bridge (MMSB). Authors propose Iterative Markovian Factorized Fitting (IMFF): alternating projections between a Markovian class and a factorized reciprocal class that mixes independent local bridges; jointly learns forward/backward dynamics.
The authors showed static–dynamic equivalence, existence/Markovianity; Theorem 3.2 positions the IMFF fixed point $\mathbb{P}^\star$ between MMSB and a pairwise SB concatenation. And illustrate state-of-the-art on 100D Embryoid Body; scalable image results up to 7 marginals with coherent unpaired video generation.

**Strengths:**

* **S1 — Principled, scalable framework:** The factorized reciprocal class + alternating projections is a clean extension of IMF to multi-marginal, enabling tractable training at scale.

* **S2 — Strong empirical reach:** SOTA on 100D biology; compelling unpaired video sequences in high dimension.

* **S3 — Sound theoretical scaffolding:** Clear static–dynamic link and KL sandwich bound; practical objectives derived from bridge conditionals; probability-flow ODE sampling avoids score estimation.

* **S4 — Solid engineering:** Fully vectorized training across intervals with masked time discretization makes the method usable in practice.

**Weaknesses:**

* **W1 — Theory gaps called assumptions:** Key steps rely on **Conjecture 3.1** and on a **variational characterization in the factorized setting** (labeled as Prop. 3.6 but effectively an assumption). Theorems that depend on these should be clearly separated.

* **W2 — Efficiency unquantified:** Inner-loop SDE simulations in IMFF (both directions) are a likely bottleneck; no big-O in $d$, no step-sensitivity, and no wall-clock comparisons vs baselines (e.g., DMSB).

* **W3 — Approximation gap unmeasured:** The bound places $\mathrm{KL}(\mathbb{P}^{\text{MMSB}} \| \mathbb{Q}) \le \mathrm{KL}(\mathbb{P}^\star \| \mathbb{Q}) \le \mathrm{KL}(\mathbb{P}^{\text{pair}} \| \mathbb{Q})$, but there is no empirical KL/energy gap to MMSB or to the pairwise construction.

* **W4 - Missing literature cited:** A few relevant papers in Mult-Marginal Flow Matching published in 2025 are not cited, I suggest a more comprehensive review of the recent literature.

* **Minor:** Notation consistency ($N$ vs $K$); briefly tighten the proof text around Theorem 3.2 to avoid implying equality with MMSB without the stated assumptions.

**Questions:**

1. **Computational efficiency (W2):**

   * How does the inner-loop SDE simulation cost scale with data dimension $d$ and number of marginals $K$?
   * What are typical integration step counts per interval, and how sensitive is stability/performance to this choice?
   * Please report **wall-clock** training times and GPU memory vs **DMSB** and other baselines in Table 2.

2. **Theoretical assumptions (W1):**

   * What are the main obstacles to proving Conjecture 3.1 and the factorized variational characterization?
   * Can you identify regimes (e.g., degenerate noise, heavy-tailed marginals, non-Gaussian reference) where these assumptions are expected to fail?
   * Given the reliance, would you relabel **Prop. 3.6** as an **Assumption** and mark all downstream results that depend on it?

3. **Approximation gap (W3):**

   * On Gaussian tests (2D and 50D), can you report **KL** or **energy** values for $\mathbb{P}^{\text{pair}}$, $\mathbb{P}^\star$, and the theoretical MMSB target (when computable or via tight bounds)?
   * Any ablation showing how gap metrics vary with number of IMFF iterations and SDE step counts?

4. **Implementation stability:**

   * Why does sequential interval training cause "path forgetting"? Is it due to optimizer drift, data imbalance across intervals, or non-stationary targets?
   * What exact aspect of the fully vectorized + masked scheme (and time encodings) stabilizes learning—e.g., synchronized gradient signals, regularization across intervals, or reduced variance?
   * Could you share an ablation (sequential vs vectorized) on a medium-size setup to quantify the stability/performance delta?RetryTo run code, enable code execution and file creation in Settings > Capabilities.Incognito chats aren’t saved to history or used to train models.

---

> ### Author Response · Authors · 2025-11-27
> **Response 1 to Reviewer wGJR**
>
> Thank you for the detailed review of our paper.
> As for the other reviewers, in this first response we start answering to some of the weaknesses and questions raised specifically in your review, additionally to a global response above.
>
> > W2 — Efficiency unquantified: Inner-loop SDE simulations in IMFF (both directions) are a likely bottleneck; no big-O in d, no step-sensitivity, and no wall-clock comparisons vs baselines (e.g., DMSB).
>
> Our framework indeed needs a simulation phase; note however that we typically use 10-20k inner iterations (optimisation steps) on a single simulation cache. With this number of inner iterations, the SDE simulation is actually not a bottleneck.
>
> > W3 — Approximation gap unmeasured: The bound places $\mathrm{KL}\left(\mathbb{P}^{\mathrm{MMSB}} \mid \mathbb{Q}\right) \leq \mathrm{KL}\left(\mathbb{P}^{\star} \mid \mathbb{Q}\right) \leq \mathrm{KL}\left(\mathbb{P}^{\text {pair }} \mid \mathbb{Q}\right) $, but there is no empirical KL/energy gap to MMSB or to the pairwise construction.
>
> We have changed Theorem 3.2 to actually take into account Conjecture 3.1.
>
> We cannot concretely measure empirical KL or energy gap wrt the true theoretical solution of MMSB, as it remains unknown (even in the Gaussian case!).
>
> We report the energy gap wrt learned pairwise SB in 4.2, but this is subject to actual model capacity and training (we stress that we did not try to maxout this metric in 4.2 and were simply happy with the fact that we were not too far away from the strong DSBM baseline).
>
> > W4 - Missing literature cited: A few relevant papers in Mult-Marginal Flow Matching published in 2025 are not cited, I suggest a more comprehensive review of the recent literature.
>
> We have updated and are still updating our literature review, as we are trying to include more baselines.
>
> > Minor: Notation consistency (N vs K); briefly tighten the proof text around Theorem 3.2 to avoid implying equality with MMSB without the stated assumptions.
>
> We indeed swapped indices mid-paper, sorry for that – it’s fixed.
>
> > How does the inner-loop SDE simulation cost scale with data dimension and number of marginals ?
>
> Our algorithm in its entirety is actually theoretically of *constant complexity* with the number of marginals! This is because of its fully parallelized nature. We only have an empirical scaling in the needed batch size dimension (we ideally want data from all marginals to be in each batch, but this isn’t even a hard constraint), and total number of simulation timesteps (which would typically be expected to scale linearly with the number of marginals).
>
> Data dimension is really just a question of model scaling; we demonstrated the nice scaling of the method in the image experiments.
>
> > What are typical integration step counts per interval, and how sensitive is stability/performance to this choice?
>
> We typically use a few tens of integration steps per interval (up to 100 in image experiments). We unfortunately did not perform extensive sweeps on this hyperparameter but expect to follow the same response as DSBM (see Figure 17 of their paper).
>
> Given that we did not need to optimize these hyperparameters for our method to converge we assume that it is stable wrt them.
>
> > Can you identify regimes (e.g., degenerate noise, heavy-tailed marginals, non-Gaussian reference) where these assumptions are expected to fail?
>
> What assumptions exactly are you referring to? We generally rely on the implicit assumption that Q is integrable (see [Leonard’s 2012 survey](https://arxiv.org/abs/1308.0215)), so heavy-tail reference processes would not fit in our current theoretical framework.
> The theoretical part is independent of whether $\mathbb{Q}$ is Brownian though.
>
> > On Gaussian tests (2D and 50D), can you report KL or energy values for $\mathbb{P}^\text{pair}$, $\mathbb{P}^\star$, and the theoretical MMSB target (when computable or via tight bounds)?
>
> We did report the energy values of our learned model in 4.2 (2D), and compared them to the ideal reference of those reported by DSBM.
>
> > Why does sequential interval training cause "path forgetting"? Is it due to optimizer drift, data imbalance across intervals, or non-stationary targets?
>
> We hypothesize that it’s due to the “optimizer drift” (concretely, the optimization objective changing along an epoch) as we generally have balanced data along marginals.
>
> > What exact aspect of the fully vectorized + masked scheme (and time encodings) stabilizes learning—e.g., synchronized gradient signals, regularization across intervals, or reduced variance?
>
> Similarly as above, we believe the synchronized gradient signal to be the main contributor.

---

> ### Author Response · Authors · 2025-12-03
> **Response 2 to Reviewer wGJR**
>
> > Given the reliance, would you relabel Prop. 3.6 as an Assumption and mark all downstream results that depend on it?
>
> We have now demonstrated Proposition 3.6.
>
> > Please report wall-clock training times and GPU memory vs DMSB and other baselines in Table 2.
>
> We report here wall-clock comparisons with DMSB:
>
> |                       | DSBM (Chen et al., 2023) |          | MMtSBM (ours) |          |
> |-----------------------|---------------------------|----------|----------------|----------|
> | Number of marginals  | 4                         | 5        | 4              | 5        |
> | Train time           | 33 min                    | 44 min   | 20 min         | 32 min   |
> | Sampling time        | 2.00 s                    | 2.02 s   | 2.00 s         | 2.00 s   |
>
>
> it is included in the annex as Table 5, along Table 7 which demonstrates the efficiency of MMtSBM up the the video domain:
>
> | Dataset  | Biotine |
> |----------|---------|
> | Number of marginals | 6 |
> | Dimension          | 128 $\times$ 128 $\times$ 3 = 49,152 |
> | Training time       | 5 h |
> | Number of epochs    | 5 |
> | Sampling time       | 32 s |
> | Generated frames    | 602 |

---

### Official Review · Reviewer_Pj4n · 2025-10-31

**Soundness:** 2
**Presentation:** 2
**Contribution:** 1
**Rating:** 4
**Confidence:** 4

**Summary:**

This paper proposes Multi-Marginal Temporal Schrödinger Bridge Matching (MMtSBM), an algorithm for trajectory inference under multi(-time-point) marginal constraints. Unlike conventional Schrödinger Bridge (SB) formulations that only match the distributions at the initial and terminal times, MMtSBM extends the framework to accommodate observations or constraints at multiple intermediate time steps. The authors establish that both reciprocal and Markovian properties remain valid in this multi-marginal setting, ensuring theoretical consistency. Building on this foundation, they develop an Iterative Markovian Fitting (IMF) algorithm to efficiently solve the resulting optimization problem. The proposed approach is evaluated on synthetic datasets and real-world single-cell RNA sequencing data.

**Strengths:**

- The paper is clearly written and easy to follow, presenting the motivation and mathematical formulation in a coherent manner.

- The authors provide a theoretically sound analysis of how multi-time-point marginal constraints can be incorporated into the SB framework while preserving the gluing dynamics and Markovian structure.

- The paper includes a range of experiments, covering both synthetic toy problems and biologically relevant datasets, demonstrating potential applicability across domains.

**Weaknesses:**

- The title and terminology could be misleading. The phrase “video generation” may create confusion, as the experiments mainly involve synthetic datasets (e.g., MNIST digit morphing or Biotin simulations) rather than natural video generation tasks. Clarifying this distinction (or changing title) would improve presentation and avoid misinterpretation.

- The theoretical novelty appears limited. The proposed extension only requires to show sort of "continuity" across time points with marginal constraints, which alone may not be a new theoretical formulation beyond the standard SB framework. From an algorithmic perspective, the proposed method resembles running multiple instances of DSBM in parallel rather than introducing a new optimization scheme.

- The experimental evaluation relies heavily on synthetic data and lacks strong baselines. For the single-cell RNA sequencing experiments, several relevant benchmarks are missing (e.g., [1, 2, 3, 6]). In addition, comparisons with concurrent works such as [4] and [5], both of which also address multi-marginal SB matching, would provide a clearer understanding of the relative advantages of the proposed approach. In particular, it would be interesting to compare whether the momentum-based formulations in [4] outperform or complement the IMF-based gluing dynamics proposed in this paper.


References

[1] A Computational Framework for Solving Wasserstein Lagrangian Flows.

[2] Multimodal Single-Cell Data Integration Challenge: Results and Lessons Learned.

[3] Simulation-Free Schrödinger Bridges via Score and Flow Matching.

[4] Momentum Multi-Marginal Schrödinger Bridge Matching.

[5] Multi-Marginal Schrödinger Bridge Matching.

[6] Multi-Marginal Schrödinger Bridges with Iterative Reference Refinement.

**Questions:**

- For the Biotin experiments, is there a quantitative baseline or evaluation metric used for comparison? A detailed explanation of the evaluation methodology would strengthen the experimental section.

- In Section 4.1, there are place holder "[CITE]".

---

> ### Author Response · Authors · 2025-11-24
> **Response 1 to Reviewer Pj4n**
>
> First and foremost, thank you for your time and insightful remarks.
> As for other reviewers, in this response we answer to each weakness and question raised specifically here, additionally to a global response.
>
> > The title and terminology could be misleading. The phrase “video generation” may create confusion, as the experiments mainly involve synthetic datasets (e.g., MNIST digit morphing or Biotin simulations) rather than natural video generation tasks. Clarifying this distinction (or changing title) would improve presentation and avoid misinterpretation.
>
> Both MNIST and biotine are real datasets (MNIST was built from real handwritten digits, while biotine is an actual cell culture experiment, as stated in 4.5.2). We thought of using this term in the title as we wanted to highlight the capability of our method to scale, for the first time, to very-high dimensional image data, yielding temporally coherent videos.
>
> We actually claim that one can hardly use a more “natural” video tasks that reconstructing the evolution of cells over time, and are uncomfortable with the way the word “natural” is usually understood in the community as it has mainly become a synonym for “classical to most” or even “available on the internet”.
>
> Since MMtSBM is also applicable on non-image data as we highlighted in some non-video experiments like the SOTA sc-RNA seq benchmark, we are indeed thinking about removing the "for video generation" part of the title (it's too long anyway).
>
> Update: we have updated the title.
>
> > The theoretical novelty appears limited. The proposed extension only requires to show sort of "continuity" across time points with marginal constraints, which alone may not be a new theoretical formulation beyond the standard SB framework. From an algorithmic perspective, the proposed method resembles running multiple instances of DSBM in parallel rather than introducing a new optimization scheme.
>
> Thank you for the thoughtful comment. We would like to clarify that the contribution of our work is not limited to enforcing continuity across time points, nor does MMtSBM amount to running several two-marginal SB solvers in parallel.
>
> The core theoretical novelty lies in the multi-marginal extension of the Iterative Markovian Fitting (IMF) framework, which is fundamentally different from standard two-point formulations. In MMtSBM, we introduce new multi-marginal projection steps and prove that their alternating application converges to the globally consistent solution satisfying all marginal constraints simultaneously. This theoretical result does not follow from the classic SB setting and is not obtained by gluing together local solutions: it is a global solution to the full multi-marginal SB problem.
>
> Algorithmically, MMtSBM is also not equivalent to parallel DSBM. It relies on a single global control shared across all intervals, whose continuity and boundary-matching properties arise directly from our theoretical construction. This global parameterization forces all local bridges to be jointly consistent, eliminating the error accumulation and trajectory discontinuities characteristic of naive sequential or parallel schemes. We stress that a naive adaptation of IMF fails to converge at all on this multi-marginal setting.
>
> In summary, MMtSBM introduces both a non-trivial theoretical extension of IMF to the multi-marginal setting and a unified optimization procedure that cannot be reproduced by running multiple independent two-marginal SB solvers.

---

> ### Author Response · Authors · 2025-11-24
> **Response 2 to Reviewer Pj4n**
>
> > The experimental evaluation relies heavily on synthetic data and lacks strong baselines. For the single-cell RNA sequencing experiments, several relevant benchmarks are missing (e.g., [1, 2, 3, 6]). In addition, comparisons with concurrent works such as [4] and [5], both of which also address multi-marginal SB matching, would provide a clearer understanding of the relative advantages of the proposed approach. In particular, it would be interesting to compare whether the momentum-based formulations in [4] outperform or complement the IMF-based gluing dynamics proposed in this paper.
>
> Firstly, we stress that neither the Transcriptomic benchmark (4.4), nor the MNIST or biotine experiments are synthetic data: all of these 3 experiments (over 6 in total) are real data, 2 of them having a true biological application.
>
> We thank the reviewer for the proposed references. These were not included because:
> - [5] was actually publicly released after us (Oct 18th on arxiv vs Oct 2 for us) which explains why it was logically not included in the comparison. We note with interest however that they came up with essentially the same idea as us!
> - [4] does not seem to use the same experimental setup as other available codebases for the EB transcriptomic benchmark, resulting in completely different evaluation values (for example compare the DMSB values in the DMSB paper and in [4]). Since they do not release any code, we unfortunately cannot compare to them because we cannot reproduce their evaluation setting. We might try to use a different data scaling pipeline to see if reported figures from DMSB for example match their reported figures. We furthermore understand anyway from the paper that they are only using analytical solutions with a strong B-spline prior, which obviously will not scale to our setting.
> - [2] we are currently trying to include the CITE and/or MULTI benchmarks.
> - [3] only reports for the 5D EB transcriptomic benchmark.
> - [1] only reports for the 5D EB transcriptomic benchmark.
> - [6] seems to be below DMSB on the EB benchmark, and they only report for the 5D EB transcriptomic benchmark.
>
> We stress that we spent quite some time making our method scale to very high dimension and image data which none of the cited papers do. We also believe that the baseline we chose (DMSB) was the SOTA before us.
>
> That being said, we totally understand that providing more comparisons is beneficial to the paper and we are currently trying to include CITE and/or MULTI benchmarks which are indeed reported in some of these references.
>
> > For the Biotin experiments, is there a quantitative baseline or evaluation metric used for comparison? A detailed explanation of the evaluation methodology would strengthen the experimental section.
>
> There is no standard quantitative evaluation for the biotine experiment as of now, as reconstructing a temporally coherent video from purely unpaired samples has never been achieved. As answered to another reviewer, we instead evaluate the method through qualitative biological consistency: preservation of spatial structure, correct fluorescence decay patterns, and realistic cytoplasmic dynamics, which are all observed in the generated videos. We will make this qualitative evaluation clearer in the revised manuscript.
>
> We are currently computing FIDs on generated videos to provide a quantitative evaluation for the biotine experiment.
>
> > In Section 4.1, there are place holder "[CITE]".
>
> We apologize for the oversight: the placeholder “[CITE]” is already replaced with the appropriate references in the updated version.

---

### Official Review · Reviewer_a4Fi · 2025-11-01

**Soundness:** 2
**Presentation:** 1
**Contribution:** 1
**Rating:** 0
**Confidence:** 3

**Summary:**

The paper extends the Schrödinger bridge problem, which connects two probability measures, to a multi-marginal setting, where one seeks a stochastic process that additionally passes through a predefined set of intermediate measures.

**Strengths:**

Paper shows superior performance of the proposed method on Trajectory Net benchmark.

The authors provided generated videos produced by their method in the supplementary material.

**Weaknesses:**

Theoretical proofs contain incorrect implications, for example in the proof of Proof of Proposition 3.1. (line 839), the feasible set A is not closed under the weak topology because absolute continuity is not in general preserved under weak convergence (one can consider normal distributions with decreasing standard deviations which converge weakly to delta measure, which is not absolutely continuous w. r. t. Lebesgue measure). This statement in the proof requires additional assumptions which are not listed in the theorem statement. It is recommended to formulate the statement of the theorem rigorously in the supplementary material, while in the main text a shortened version could be introduced.

The paper does not clearly show the advantage of multimarginal formulation in comparison to a sequential application of Schrödinger bridge. The experiments with gaussians and cell movements don't show specific scenarios when the proposed method is superior to sequential application of Schrödinger bridges. For example, does the proposed methods is better that sequential application of Schrödinger bridges if all marginals lie on the same Wasserstein geodesic?

Despite video generation being claimed in the abstract as one of the main contributions of the paper, it is evaluated only on a single video dataset. Video-specific datasets are not considered. Moreover, the provided videos appear too similar to simple image blending, making it difficult to assess whether the method can learn coherent motion from unpaired frame data. The paper lacks an experiment on a conventional real video dataset, from which an unpaired frame dataset could be produced, allowing the quality of generated videos to be evaluated using the Fréchet Video Distance or some optical flow-based metric for motion quality estimation.

The text of the paper is poorly written and difficult to follow. It contains too many generic phases, particularly in the abstract. (for example phrases like “in an efficient and principled way” in line 064). The paper contains missed citations “it is a pure translation of each Gaussian component inside the mixtures [CITE]. After only the warm-up phase (akin to flow matching [CITE], as said before),” in line 335.

​​Experiment 4.3 “Since no closed-form solution is available for the static multi-marginal SB”. There is a closed-form solution for entropic optimal transport between gaussians of any dimension for pairwise (sequential) optimal transport [1], but it wasn’t used for evaluation, as far as the reviewer understood from the paper.

The multimarginal optimal transport is a complex problem and optimal couplings may have a complex or even fractal form [2]. However it cannot typically be reduced to the pairwise costs like in line 136. This raises the question on how the proposed method behaves for different transport costs. Does it work for costs like c(x,y,z) = x y z or is it limited to cost of the form c(x,y,z) = (x-y)^{2}+(y-z)^{2} (i.e. when there is a sequential separation of costs like c(x,y,z,t,v) = f(x,y)+f(y,z)+f(z,t)+f(t,v))

Methods like Action Matching [3] are also capable of reconstructing dynamics from a sequence of marginal distributions. However such methods were not considered in evaluation.

Generated videos provided in the supplementary page look like a blending between images and do not recover the cell movements as one would expect. More importantly, there are no comparisons in the supplementary with other methods for the video generation task. This questions the contribution stated in the abstract "for the first time recovers couplings and dynamics in very high dimensional image settings".

[1] Hicham Janati, et. al., Entropic Optimal Transport between Unbalanced Gaussian Measures has a Closed Form.

[2] Gladkov et. al., On multistochastic Monge–Kantorovich problem, bitwise operations, and fractals

[3]  Neklyudov et. al., Action Matching: Learning Stochastic Dynamics from Samples

**Questions:**

How does the proposed method differ from the sequential application of the Schrödinger bridge?

In line 097: "In the limit ε → 0, this recovers classical OT, which motivates our interpolation framework." How the convergence of the entropy regularised optimal transport to the classical OT problem motivates the proposed method?

**Details Of Ethics Concerns:**

There are no ethnics concerns.

---

> ### Author Response · Authors · 2025-11-24
> **Response 1 to Reviewer a4Fi**
>
> We thank the reviewer for their very detailed comments.
> In this response we answer to each weakness and question raised specifically here, additionally to a global response.
>
> > Theoretical proofs contain incorrect implications, for example in the proof of Proof of Proposition 3.1. (line 839), the feasible set A is not closed under the weak topology because absolute continuity is not in general preserved under weak convergence (one can consider normal distributions with decreasing standard deviations which converge weakly to delta measure, which is not absolutely continuous w. r. t. Lebesgue measure). This statement in the proof requires additional assumptions which are not listed in the theorem statement.
>
> You are completely right that the admissible set is in general not weakly closed –we apologize for this wrong statement in the Appendix.
> Fortunately, this does not affect Proposition 3.1 as we do not, actually, need the feasible set to be closed: the first argument about feasibility of the problem relies in reality on the convexity of the set *only* (with the strict convexity of $P \mapsto KL(P\|Q)$), independently of any closeness assumption. We have corrected the statement accordingly in the revised version.
>
>
> > The paper does not clearly show the advantage of multimarginal formulation in comparison to a sequential application of Schrödinger bridge. The experiments with gaussians and cell movements don't show specific scenarios when the proposed method is superior to sequential application of Schrödinger bridges. For example, does the proposed methods is better that sequential application of Schrödinger bridges if all marginals lie on the same Wasserstein geodesic?
>
> Compared to the sequential learning of distinct models, the advantages of our approach are both theoretical and practical.
>
> The first practical advantage is actually common to the entire multi-marginal literature and not specific to our method: it is to train a single bridge model (so 2 nets: one for each direction) instead of $n$ models ($2n$ nets). This naive linear scaling of bridges renders tasks with many marginals quite impractical to train concretely.
>
> Another –theoretical– advantage is to ensure that each section of the bridge is consistent with the former or later one at any stage of the training, contrary to sequentially training distinct models. Indeed, even in a theoretical regime of perfect end convergence of such sequential models, nothing would force each bridge to match the outputs of the former generation during training, resulting in incoherent full trajectory generation of any non-perfect model, which is of course always the case in reality. In our case, the parallel Gyöngy projection applied simultaneously to all time indices ensures that the entire set of intermediate drifts collapses to a single forward drift and a single backward drift, consistent across all times.
>
> Finally, even if a single conditional model is trained like in our case, another practical advantage of our method compared to sequentially learning the bridges is the important avoidance of model forgetting, which is actually one of the key tricks making our method successful. Indeed, although we do not include this experiment in the appendix, we quickly realized that naively applying classical SB training sequentially on each pairwise bridge produces completely inconsistent drift fields because the model cannot generalize anymore to previous times as it has “forgotten” them.
> We note that we interpret for example the concrete algorithm proposed in [DMSB](https://arxiv.org/abs/2303.01751) as trying to compensate for this inconsistency by introducing an artificial iterative projection scheme: they repeatedly re-apply their projection and even condition directly from the first to the last marginal as an additional regularization step. This has no clear theoretical justification and mainly compensates for the incompatibility created by sequential SBs.

---

> ### Author Response · Authors · 2025-11-24
> **Response 2 to Reviewer a4Fi**
>
> > Despite video generation being claimed in the abstract as one of the main contributions of the paper, it is evaluated only on a single video dataset. Video-specific datasets are not considered. Moreover, the provided videos appear too similar to simple image blending, making it difficult to assess whether the method can learn coherent motion from unpaired frame data. The paper lacks an experiment on a conventional real video dataset, from which an unpaired frame dataset could be produced, allowing the quality of generated videos to be evaluated using the Fréchet Video Distance or some optical flow-based metric for motion quality estimation.
>
> We evaluate our method on 2 “video” datasets: MNIST (admittedly low dimensional but still images –hence videos) and biotine. We fully agree with you that more video experiments and evaluations would be beneficial.
>
> We stress that video generation from purely unpaired frames is totally new, so no benchmarks exist at the present time. Because of this lack of baselines we believe raw metrics like FVD to be hardly interpretable, and the comparison with models training on videos obviously unfair.
>
> We are still very confident with “whether the method can learn coherent motion from unpaired frame data” because of the nice motion we see on the MNIST frames.
>
> > The text of the paper is poorly written and difficult to follow. It contains too many generic phases, particularly in the abstract. (for example phrases like “in an efficient and principled way” in line 064). The paper contains missed citations “it is a pure translation of each Gaussian component inside the mixtures [CITE]. After only the warm-up phase (akin to flow matching [CITE], as said before),” in line 335.
>
> We are sorry that our writing felt poor to you and made a number of changes to improve the paper; they are listed in the global response.
>
> We believe our method to be 1) efficient and 2) principled in the way that 1) the learning algorithm is fully parallelized across all times, in stark comparison for example with [DMSB](https://arxiv.org/abs/2303.01751) who treats marginals in an iterative way, and 2) we have demonstrated the mathematical soundness of our algorithm and presented our theoretical results using the presentation of [DSBM](https://arxiv.org/abs/2303.16852) which we believe to be particularly sound.
>
> We apologize for the missing citations and we have fixed this unfortunate typo in the revised manuscript.
>
> > ​​Experiment 4.3 “Since no closed-form solution is available for the static multi-marginal SB”. There is a closed-form solution for entropic optimal transport between gaussians of any dimension for pairwise (sequential) optimal transport [1], but it wasn’t used for evaluation, as far as the reviewer understood from the paper.
>
> It is indeed what we have used as comparison in Experiment 4.3 (“Since no closed-form solution is available for the static multi-marginal SB, we compare our method to the sequence of theoretical results for each pairwise SB (Bunne et al., 2023).“).
>
> > The multimarginal optimal transport is a complex problem and optimal couplings may have a complex or even fractal form [2]. However it cannot typically be reduced to the pairwise costs like in line 136. This raises the question on how the proposed method behaves for different transport costs. Does it work for costs like c(x,y,z) = x y z or is it limited to cost of the form c(x,y,z) = (x-y)^{2}+(y-z)^{2} (i.e. when there is a sequential separation of costs like c(x,y,z,t,v) = f(x,y)+f(y,z)+f(z,t)+f(t,v))
>
> We stress that all theoretical results, including the convergence of the algorithm, do not rely on any particular form of the reference process $\mathbb{Q}$ and thus hold for any associated cost structure (*eg* in [Léonard 2012](https://arxiv.org/abs/1308.0215), the cost is not fixed to be quadratic: any reference process satisfying a suitable large-deviation principle generates its own rate functional and therefore induces a different optimal transport cost).
>
> We indeed use a Brownian motion for all experiments because of its simplicity, its apparent effectiveness w.r.t. tasks that we benchmark MMtSBM against, the lack of a good prior, and because of its universality in the literature.
>
> Exploring other reference processes (and thus associated OT costs) is definitively a very interesting line of research that we would like to explore in future works.

---

> ### Author Response · Authors · 2025-11-24
> **Response 3 to Reviewer a4Fi**
>
> > Methods like Action Matching [3] are also capable of reconstructing dynamics from a sequence of marginal distributions. However such methods were not considered in evaluation.
>
> There is indeed a developed literature of methods "reconstructing dynamics from a sequence of marginal distributions", that we of course compare against, since this is exactly what we do. We actually report state-of-the-art scores on the now classical and widely used 100D Embryoid (EB) scRNA-Seq Data (proposed as benchmark in [TrajectoryNet](https://arxiv.org/abs/2002.04461)).
>
> Regarding [Action Matching](https://arxiv.org/abs/2210.06662), we note that they only report on the 5D version of the EB benchmark, and claim the following: “We find that eAM [their method] performs *on par with NLSB* and outperforms other methods.” (emphasize ours).
>
> We would like to bring to the attention of the reviewer that NLSB is the worst performing method of Table 2 (on which we are SOTA), and that we outperform reported metrics of NLSB by a factor of 4 in terms of SWD, and of 33 in terms of MMD on this task, while using a network of 300k parameters vs 1.3M for example for [DMSB](https://arxiv.org/abs/2303.01751), the second-best method after ours.
>
> Globally, apart from the initial verifications on toy examples, we are interested in scaling our method to high dimensions and are thus focusing on high-dimensional experiments and benchmarks.
>
> > Generated videos provided in the supplementary page look like a blending between images and do not recover the cell movements as one would expect.
>
> This observation is very interesting and actually *expected* on this dataset.
>
> As we can see on the MNIST experiment, MMtSBM is fully capable of rendering important changes in pixel space (indeed, removing or adding fully white regions on the image to form a number yields a rather massive change w.r.t. the previous image).
>
> Yet, a mostly static *positional* evolution is indeed observed on the biotine dataset. This actually stems from the fact that cell position is not a statistically varying information on the biotine dataset, and this (non-existing) signal is thus simply not seen by our purely unpaired method, resulting in non-moving cells. MMtSBM rather reconstructs the OT trajectory in pixel space, yielding indeed very close cellular position while still accurately matching the time-varying phenotype (for example, we clearly observe fluorescence loss in the cytoplasmic area, along with other expected behaviors).
>
> Rather than artificially moving cells around, MMtSBM produces a temporally consistent video that matches the scientifically-pertinent signal of this dataset.
>
> > More importantly, there are no comparisons in the supplementary with other methods for the video generation task. This questions the contribution stated in the abstract "for the first time recovers couplings and dynamics in very high dimensional image settings".
>
> The reason we do not include comparisons with existing video generation methods is that our setting is fundamentally different from standard video generation benchmarks. We do not perform video prediction or video synthesis from paired temporal training data (ie videos). Instead, we address a much more constrained problem: recovering biological dynamics from unpaired static images, where the model has never seen matched time sequences during training. Classical video generation models (e.g., diffusion-based video generators, autoregressive video models, or video GANs) require paired frames, dense temporal supervision, and large-scale video datasets, which makes them inapplicable to the fully unpaired setting.
>
> Our contribution is precisely to show that it is possible, for the first time, to reconstruct meaningful dynamics in high-dimensional image settings without temporal supervision and solely from static marginal distributions. Since no existing approach can be applied to this specific unpaired scenario, standard quantitative comparisons are not feasible. Instead, we evaluate the method through qualitative biological consistency: preservation of spatial structure, correct fluorescence decay patterns, and realistic cytoplasmic dynamics, which are all observed in the generated videos.

---

> ### Author Response · Authors · 2025-11-24
> **Response 4 to Reviewer a4Fi**
>
> > How does the proposed method differ from the sequential application of the Schrödinger bridge?
>
> The “sequential application of the Schrödinger bridge” can be achieved in 2 ways:
> - either learning n independent bridges (2n networks) which is clearly impracticable as the number of marginals grow (and, maybe even most importantly, very inelegant…)
> - or by naively learning a single network by looping over batches of a single marginal pair, which simply fails to train (note that [DMSB](https://arxiv.org/abs/2303.01751) does learn the full multi-marginal bridge but adds additional start-to-end regularizations in their actual algorithm that have no theoretical grounding –we find in some experiments that this algorithm actually fails to converge at all)
>
> MMtSBM differs by:
> - defining the multi-marginal temporal Schrödinger Bridge problem
> - demonstrating its fundamental theoretical properties
> - introducing a novel factorized extension of the IMF algorithm for multiple marginals
> - producing a theoretical convergence analysis of the algorithm under asymptotic hypotheses
> - proposing to learn the bridge not sequentially over the total SB but rather in parallel over all marginal pairs of the full trajectory, along with other important considerations (see Appendix A.4)
>
> > In line 097: "In the limit ε → 0, this recovers classical OT, which motivates our interpolation framework." How the convergence of the entropy regularised optimal transport to the classical OT problem motivates the proposed method?
>
> Thank you for the question. Our reference to the limit $\epsilon \rightarrow 0$ is not used as a technical ingredient of our algorithm, but as a modeling intuition. In this limit, the multi-marginal Schrödinger Bridge converges to multi-marginal quadratic OT, which selects the minimum-displacement coupling between successive marginals. This describes the idea of a natural or energy-efficient evolution between distributions.
>
> This intuition fits our application: we aim to learn videos or dynamics between unpaired images, and without temporal supervision a reasonable prior is that a plausible video should evolve by minimizing pairwise transport cost at each time step. In other words, among all feasible transitions, the one that changes the image the least, according to the geometry of quadratic OT, is the most natural.
>
> Working with the entropic formulation ($\epsilon > 0$) provides a smooth and tractable version of this principle while still reflecting the minimal-displacement structure of OT. It is therefore a model and working hypothesis that guides the design of our method.

---

### Author Response · Authors · 2025-11-24
**First global comment to all reviewers**

Dear reviewers,

Firstly, we would like to thank you all again for the precious time that you put into reviewing our article.

As we are gradually answering you point by point in dedicated responses, we would like to inform you that we already made a number of revisions to our manuscript, taking into account some of your remarks already.

Namely:
- we fixed typos and dangling citations / missing variable definitions, sorry again for that...
- we improved Table 1 in 4.2 and made it clear all other metrics were from DSBM directly, thus yielding a strong ideal vanilla SB baseline
- we explained better the warmup phase that we perform, making the interest of experiment 4.1 (concretely: an ablation study of our algorithm) much clearer
- we improved the readability of the paper in various places, especially concerning figures
- we checked that our intuition in 4.2 ("In this configuration the optimal transport between each pair of marginals is known exactly: it is a pure translation of each Gaussian components inside the mixture") was indeed correct with an empirical yet exact OT solver: POT.
- we added some of the references you suggested to the introduction
- we better explained the qualitative evaluation of the biotine cell culture experiment
- we corrected the proof of proposition 3.1
- we added uncertainty estimations to Table 2, confirming that our SOTA results hold with a wide margin

We are now focusing on finishing all our individual responses and improving the evaluations of our method.

Thank you very much for your valuable feedback,

The authors.

---

### Author Response · Authors · 2025-12-03
**Second global comment to all reviewers**

Dear ACs and (unfortunately muted) reviewers,

This response is the continuation of the previous “First global comment to all reviewers”

We have additionally:

- added an entirely *new and widely reported benchmark on which we also achieve statistically significant SOTA scores* against comparable methods: the leave-one-time-out 100D MULTI dataset; this allows us to address one of the key issues raised by reviewers: lack of comparison to more recent papers than DMSB; we are thus more confident than ever that our method is strongly competitive for singe-cell trajectory inference despite lacking specialized modeling or ad-hoc priors

| Method                     | 𝒲₁ (↓)            |
|----------------------------|--------------------|
| **Schrödinger Bridge**     |                    |
| WLF-SB                     | 55.065 ± 5.499     |
| [SF]²M-Exact               | 52.888 ± 1.986     |
| [SF]²M-Geo                 | 52.203 ± 1.957     |
| __MMtSBM__                 | __44.542 ± 0.637__ |
| **No precomputed OT conditioning** |            |
| I-CFM                      | 57.262 ± 3.855     |
| I-MFM\_RBF                 | 54.197 ± 1.408     |
| __MMtSBM__                 | __44.542 ± 0.637__ |

---

| Method                     | 𝒲₁ (↓)            |
|----------------------------|--------------------|
| **Wasserstein Gradient Flows** |                |
| WLF-SB                     | 55.065 ± 5.499     |
| WLF-OT                     | 55.416 ± 6.097     |
| WLF-UOT                    | 54.222 ± 5.827     |
| WLF-(OT+potential)         | 47.365 ± 0.051     |
| WLF-(UOT+potential)        | 45.231 ± 0.010     |
| **Flow Matching with exact OT conditioning** |  |
| OT-CFM                     | 54.814 ± 5.858     |
| OT-MFM\_RBF                | 50.906 ± 4.627     |
| **Metric-aware *interpolation* with exact OT conditioning** | |
| GAGA                   | 27.04 ± 2.95   |

- proved proposition 3.6, making our theoretical contributions much more robust
- added quantitative metrics for the biotine video experiments, reporting DINOv2-KIDs; we stress that, to the best of our knowledge, no baseline exist for the task of video generation from purely unpaired data, let alone one approaching the OT plan through Schrödinger Bridge learning
- improved writing and organization in various places (eg proof of proposition 3.8 in the appendix, more literature review, fix indices swapping, etc)
- we updated the title to reflect the variety of tasks involved in our paper

We have also finished responding to individual points raised by reviewer wGJR.

We want to thank again the reviewers for their valuable feedback; we hope that it allowed us to make our paper much better,

The authors.

---

### Meta-Review · Area_Chair_Hjer · 2026-01-07

**Summary:**

This paper considers the multi-marginal Schrödinger Bridge problem, which is an important task with scope that well fits ICLR. The main contribution is the proposition of a solver that scales better with dimensionality, which is an outstanding problem. The approach is to generalize the recent technique of Diffusion Schrödinger Bridge Matching to a multi-marginal setting. The majority of reviewers and I agree that this is an interesting idea. However, major concerns about technical novelty, presentation, experiments, and insufficient comparison with the literature were also raised. Unfortunately, these concerns seem to remain largely unresolved after the rebuttal discussion. Recognizing the potential of the idea, I encourage the authors to take the discussions into consideration and re-submit a revised version.

**Reviewer Concerns:**

Just a guess: they may not be fully convinced.

**Reviewer Scores:**

Just a guess: they may not increase by too much.

---

### Decision · Program_Chairs · 2026-01-26

Reject